# MICROMIX: EFFICIENT MIXED-PRECISION QUANTIZATION WITH MICROSCALING FORMATS FOR LARGE LANGUAGE MODELS

**Wenyuan Liu**[1], **Haoqian Meng**[1], **Yilun Luo**[1], **Peng Zhang**[1*], **Xindian Ma**[1]
[1] College of Intelligence and Computing, Tianjin University, Tianjin, China
{lwy2020, typedef, lyl2023, pzhang, xindianma } @tju.edu.cn

## ABSTRACT

Quantization significantly accelerates inference in large language models (LLMs) by replacing original high-precision matrices with low-precision counterparts. Recent advances in weight-activation quantization have primarily focused on mapping both weights and activations to the INT4 format. Although the new FP4 Tensor Cores in NVIDIA's Blackwell architecture offer up to $4\times$ speedup over FP16, existing INT4-based kernels fail to fully exploit this capability due to mismatched data formats. To bridge this gap, we propose MicroMix, a co-designed mixed-precision quantization algorithm and GEMM kernel based on Microscaling (MX) data formats. Tailored for the Blackwell architecture, the MicroMix kernel supports arbitrary combinations of MXFP4, MXFP6, and MXFP8 channels, and produces BFloat16 outputs. To achieve a favorable trade-off between accuracy and efficiency for each linear layer, we introduce quantization thresholds that identify activation elements where lower-precision formats (MXFP4 or MXFP6) incur excessive quantization error. Our algorithm selectively allocates higher-precision channels to preserve accuracy while maintaining compute efficiency. On the Llama and Qwen model families, MicroMix achieves near-FP16 performance across diverse downstream tasks with an average precision of 5 bits. In particular, Qwen2.5-32B-Base, Coder and Math exhibit lossless accuracy on zero-shot, code generation, and mathematical reasoning benchmarks. In addition, on RTX 5070Ti laptop and RTX 5090 GPUs, our kernel achieves 2.29-3.38$\times$ acceleration compared to TensorRT-FP16. Our code is available at `https://github.com/lwy2020/MicroMix`.

## 1 INTRODUCTION

In recent years, large language models (LLMs) have demonstrated remarkable performance across a wide range of tasks (Vaswani et al., 2023; Brown et al., 2020). However, these capabilities come with substantial computational and energy costs. To mitigate this, quantization techniques replace high-precision matrix multiplications with more efficient low-bit alternatives (Yao et al., 2022; Xiao et al., 2024), significantly improving LLM inference speed. Quantization techniques are broadly classified into weight-only and weight-activation approaches. Weight-only methods (Lin et al., 2024b; Frantar et al., 2023; Yang et al., 2025) have substantially mitigated the precision loss associated with 4-bit weights and 16-bit activations (W4A16). In parallel, weight-activation methods (Dettmers et al., 2022; Xiao et al., 2024) suppress activation outliers effectively, enabling accurate 8-bit quantization of both weights and activations (W8A8). More recently, mixed-precision and rotation-based quantization algorithms (Ashkboos et al., 2024; Zhao et al., 2024) have pushed the frontier further to W4A4, achieving strong performance on downstream tasks.

Despite these advances, two key bottlenecks continue to restrict the kernel-level efficiency of INT4-based quantization: (1) The widely adopted group-wise integer quantization scheme requires dequantizing each integer group to floating-point values followed by partial summations. This procedure is executed on slower CUDA Cores, as INT8 Tensor Cores only support INT32 accumulation.

---

*Corresponding Author: Peng Zhang

(2) NVIDIA's latest Blackwell architecture introduces FP4 Tensor Cores that offer up to $4\times$ higher throughput than FP16 and $2\times$ higher than FP8 or INT8. However, existing INT-based quantization kernels are incompatible with these new tensor cores and thus fail to leverage their full potential. As a result, significant room remains for optimizing quantization kernel throughput on the Blackwell architecture.

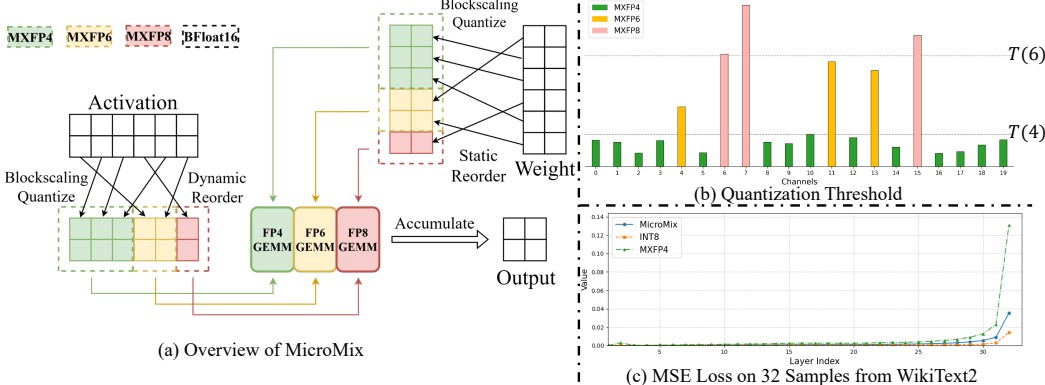

Figure 1: (a) MicroMix reorders channels and allocates different bit-widths accordingly. (b) The quantization thresholds $T(4)$ and $T(6)$ partition elements into three groups based on their quantization error magnitude. (c) MicroMix consistently achieves lower quantization error across all layers.

In this paper, we propose **MicroMix**, a mixed-precision quantization framework based on Microscaling (MX) data formats, featuring a co-designed algorithm and kernel. The key components of MicroMix are as follows:

**(1) Flexible bit-width ratios (4, 6, and 8 bits).** To balance efficiency and accuracy, MicroMix assigns customized ratios of three precision levels to each linear layer. The quantization kernel supports multiple Microscaling formats (MXFP8, MXFP6, MXFP4) and arbitrary mixing ratios. By leveraging CUTLASS GEMM, we instantiate optimized matrix multiplication kernels tailored to specific data types and problem sizes. In addition, dequantization operations are deeply fused into MMA instructions, introducing negligible overhead on Blackwell Tensor Cores.

**(2) Low-error precision assignment strategy.** We propose a bit allocation algorithm that adapts to input distributions from the perspective of quantization error. The key idea is to ensure that the quantization error of lower-bit formats remains below the upper bound of higher-precision formats. To this end, we define explicit quantization thresholds for MXFP4 and MXFP6: elements exceeding the threshold at a given bit-width are reassigned to higher-precision formats (see Figure 1(b)). This formulation introduces explicit outlier thresholds for MXFP4 and MXFP6, addressing a limitation of prior work. As a result, MicroMix significantly reduces the quantization error induced by MXFP4, as shown in Figure 1(c).

**(3) Efficient reorder-and-quantize operation.** Since adjacent channels may be assigned different bit-widths, channels of the same precision need to be reordered into the same block. Without reordering, applying mixed-precision quantization directly results in irregular memory access and considerable overhead. To address this, MicroMix integrates the reordering step into the quantization kernel (Figure 1(a)), enabling high-throughput quantization across heterogeneous precision levels with negligible additional latency.

We evaluate MicroMix on multiple downstream tasks, including zero-shot and few-shot learning, language modeling, code generation and mathematical reasoning. Across various Llama and Qwen models, MicroMix generally maintains at least 98% FP16 accuracy on zero-shot, code and math benchmarks, achieving comparable to or better than state-of-the-art baselines. In particular, MicroMix achieves near-FP16 performance on Qwen2.5-32B models (Base and Coder) with an average bits about 5.2. For efficiency analysis, we evaluate the MicroMix kernel on RTX 5070Ti laptop, RTX 5090 and RTX PRO 6000 GPUs. Compared with TensorRT-FP16, MicroMix achieves a kernel-level speedup of 2.45-2.93$\times$ on the RTX 5070Ti laptop and 2.29-3.38$\times$ on the RTX 5090. When integrated into the Transformer architecture, MicroMix achieves 1.98-2.02$\times$ higher end-to-

end compared to FP16. For end-to-end efficiency, MicroMix delivers at least 1.82 times higher decoding throughput than INT4 baselines on RTX PRO 6000.

## 2 PRELIMINARY AND MOTIVATIONS

### 2.1 PRELIMINARY

Given an activation tensor $\boldsymbol{X}$ and a weight tensor $\boldsymbol{W}$, quantization approximates the original matrix multiplication with a low-precision computation:

$$\boldsymbol{Y} = \boldsymbol{X}\boldsymbol{W} \approx Q(\boldsymbol{X})Q(\boldsymbol{W}) \cdot s_{\boldsymbol{X}}s_{\boldsymbol{W}}, \quad Q(\boldsymbol{X}) = round(\frac{\boldsymbol{X}}{s}) \tag{1}$$

where $s_{\boldsymbol{X}}$ and $s_{\boldsymbol{W}}$ are the scaling factors of $\boldsymbol{X}$ and $\boldsymbol{W}$ respectively. $\forall X_j \in \boldsymbol{X}$, the quantization error for $X_j$ is defined as:

$$E(X_j) = |X_j - Q(X_j)| = |X_j - round(\frac{X_j}{s})s| = \gamma \cdot s \tag{2}$$

where $\gamma = |round(X_j/s) - X_j/s|$ is the rounding error. In Appendix C.1, we further analyze the relationship between quantization error and model accuracy. Empirically, we observe that model accuracy remains close to the FP16 baseline as long as the quantization error is constrained within a specific threshold. However, once the quantization error exceeds this threshold, accuracy degrades rapidly. For recent LLMs, INT8 quantization typically remains within the high-accuracy region, whereas INT4 often lies near the onset of significant accuracy degradation.

Microscaling data formats (MX) are advanced numerical formats designed for deep learning. The basic unit of MX is a block of size $k$, consisting of $k$ scalar elements $\{X_j\}_{j=1}^k$ and a single shared scaling factor $s$ in E8M0 (Darvish Rouhani et al., 2023a). Recently, DeepSeek V3.1 (DeepSeek-AI, 2024) was trained using the UE8M0 FP8-scaled data format for both model weights and activations, ensuring compatibility with microscaling formats. Given a FP16 tensor $\boldsymbol{X} \in \mathbb{R}^{L \times I}$, quantization to MXFP8/MXFP6/MXFP4 first partitions $\boldsymbol{X}$ into blocks of 32 elements $\{\boldsymbol{X}_i\}_{i=1}^N, N = \frac{L \cdot I}{32}$, then applies per-block symmetric quantization for $\forall X_j \in \boldsymbol{X}_i$ as follows:

$$Q(X_j) = round(\frac{X_j}{s}), s = 2^{\lfloor \log_2(\max(|\boldsymbol{X}_i|)) \rfloor - b} \tag{3}$$

where $round(\cdot)$ denotes rounding to the nearest MXFP value and the exponent bias $b$ is format-specific (see Appendix B for more details).

### 2.2 MOTIVATIONS

The primary motivations of this paper stem from addressing the limitations in current quantization methods and their corresponding kernels.

**Motivation 1: Adaptive Mixed-precision Allocation for Diverse Activation Distributions.** Existing mixed-precision quantization methods such as Atom (Zhao et al., 2024), employ a fixed number of high-precision channels across all layers. This uniform allocation fails to account for the heterogeneous activation distributions observed in different layers (see Figure 2). Specifically, layers with larger activation values across channels require more high-precision channels to reduce the quantization error. Consequently, directly applying current fixed-allocation mixed-precision algorithms to MX formats leads to a noticeable degradation in accuracy (see Table 11 in Appendix D.3). To overcome this, we propose a novel strategy that

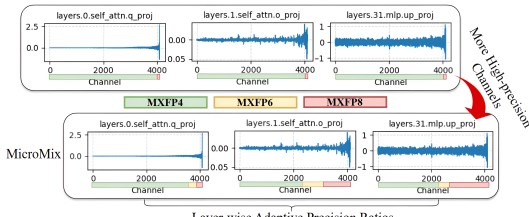

Figure 2: Channel-wise mean values of three activation tensors from Llama3.1-8B, with outlier channels reordered to the end. Compared to prior methods, MicroMix assigns a larger portion of channels to higher-precision formats and applies layer-wise adaptive precision ratios across all linear layers.

flexibly allocates the number of 4, 6, and 8-bit channels per layer. This adaptive approach ensures that all linear layers consistently maintain low errors, thereby improving model accuracy.

**Motivation 2: Leveraging FP4 Tensor Cores for Enhanced Kernel Efficiency.** Current INT-based kernels, exemplified by Atom and QuaRot (Ashkboos et al., 2024), require dequantization on CUDA Cores because INT8 Tensor Cores only produce INT32 partial sums. The dequantization process on CUDA Cores limits the performance of these INT kernels (Lin et al., 2025b). In stark contrast, FP4 matrix multiplication allows for direct dequantization on FP4 Tensor Cores, leading to a significant improvement in computational efficiency. This fundamental advantage highlights the critical need for developing next-generation mixed-precision methods with kernels specifically designed for FP formats.

**Motivation 3: Quantization Error Management through Adaptive Thresholding for Outliers.** Quantization error, inherent to the format conversion between original activations $X$ and their quantized representations $Q(X)$, cannot be entirely eliminated. Therefore, it is critical to ensure this error remains within an acceptable bound. While prior works have introduced various techniques, such as smoothing, rotation, and clipping, to mitigate the impact of outliers in activations. There is still a notable gap in research concerning the precise threshold above which outliers should be constrained for MXFP4 and MXFP6. In this paper, we define specific quantization thresholds for MX. Elements exceeding these defined thresholds will be preferentially stored in higher bit-width, thereby effectively minimizing quantization error and maintaining high model fidelity.

## 3 METHOD

To address the accuracy degradation observed in INT4 quantized models on downstream tasks, prior work has explored various solutions. However, post-training quantization for Microscaling (MX) formats remains underexplored. Leveraging the inherent flexibility of multiple MX data formats, we propose MicroMix, a novel co-designed mixed-precision quantization algorithm and kernel.

### 3.1 ALGORITHM

In MicroMix, the activation tensor channels are partitioned into three groups, $G_4$, $G_6$, and $G_8$, which are quantized to MXFP4, MXFP6, and MXFP8, respectively. The corresponding weight channels are quantized to the same bit-width as their activation counterparts.

**Reducing Quantization Error through Permutation.** Due to the limited bit-width, the quantization error of MXFP4 or MXFP6 cannot, in general, be lower than that of INT8. Given a token $X \in \mathbb{R}^I$, our key idea is to constrain the quantization error of MXFP4 and MXFP6 such that it remains within the upper bound of the error introduced by INT8:

$$E(X)_{MXFP\{4,6\}} \leq \overline{E}(X)_{INT8}, \quad \forall X \in G_{\{4,6\}} \tag{4}$$

According to Equation 2, the reduction of quantization error in MX primarily depends on lowering the maximum value within each block of $X_j$. A straightforward approach is to group large values into the same blocks while keeping smaller values together. To achieve this, we introduce a permutation $\sigma$ that rearranges the elements of $X$ in ascending order:

$$\sigma : X \rightarrow \sigma(X) \tag{5}$$

**Defining Quantization Threshold for Partitioning.** After permutation, the next step is to determine the groups $G_4, G_6, G_8$. To accurately distinguish outliers from regular elements, we define the quantization threshold as follows:

**Definition 1.** *Given a high-precision bit-width (e.g., 8-bit for recent LLMs) and a target bit-width $n$, the quantization threshold $T(n)$ is defined as:*

$$T(n) = 2^b \cdot \frac{2^{n-1}}{q_{max}} \cdot \overline{E}(X)_{INT8} \tag{6}$$

*To maintain low quantization error at MXFP4 or MXFP6, the maximum allowable magnitude within group must satisfy:*

$$\max(|G_n|) \leq T(n) \tag{7}$$

Here, $n$ denotes the number of bits, $b$ is the exponent bias and $q_{max}$ represents the maximum representable value in the target format. A detailed derivation of the quantization threshold is provided in Appendix C.2. Based on the thresholds, the groups $\boldsymbol{G}_4$, $\boldsymbol{G}_6$, and $\boldsymbol{G}_8$ are defined as:

$$\boldsymbol{G}_4 = \{X | X \leq T(4)\} \quad \boldsymbol{G}_6 = \{X | T(4) < X \leq T(6)\}, \quad \boldsymbol{G}_8 = \{X | T(6) < X\} \tag{8}$$

We calculate the proportions $p_4$, $p_6$, and $p_8$ corresponding to the channel groups $\boldsymbol{G}_4$, $\boldsymbol{G}_6$, and $\boldsymbol{G}_8$ for each linear layer in Llama3.1-8B. The results are shown in Figure 3. We summarize three key observations:

**Layer-wise Adaptivity**: The proportions vary dynamically across layers, reflecting the diverse input distributions in each activation. This demonstrates that the mixed-precision allocation is layer-specific rather than fixed globally.

**FP4 Dominance**: The proportion $p_4$ consistently exceeds 50%, indicating that FP4 computations dominate the mixed-precision workflow. This dominance contributes significantly to the computational efficiency of the model.

**Cross-Dataset Stability**: The variations of $p_4$, $p_6$, and $p_8$ across different datasets and sampling strategies are minimal, suggesting that the mixed-precision assignment remains relatively stable.

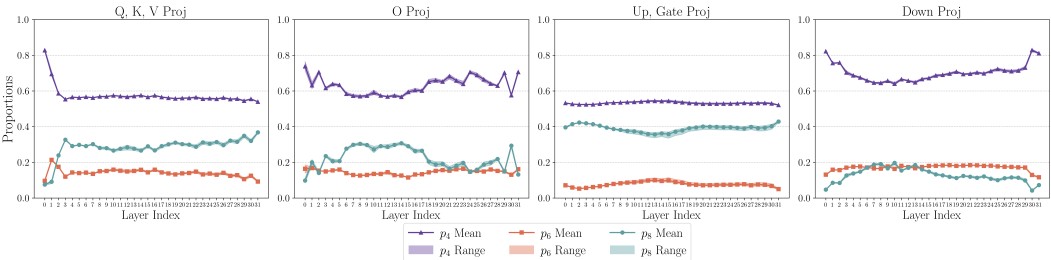

Figure 3: Distribution statistics of $p_4$ (E2M1), $p_6$ (E3M2), and $p_8$ across Llama3.1-8B. We evaluate 32 samples selected from WikiText2 (Merity et al., 2016) and the Pile dataset (Gao et al., 2020), covering batch sizes of 8, 16, 32, and 64, and sequence lengths of 512, 1024, 2048, and 4096. For each sample, $p_4$, $p_6$, and $p_8$ are computed over all linear layers. The figure reports the mean values and min-max ranges of $p_4$, $p_6$, and $p_8$ across all samples.

**Offline Channel Assignment Strategy.** Online evaluation of channel partitioning would introduce substantial runtime overhead. Instead, leveraging the observed stability of $p_4$, $p_6$, and $p_8$, we precompute $\{p_4^k, p_6^k, p_8^k, \sigma^k\}$ for the $k$th linear layer offline using calibration data. To allocate higher precision to more critical channels, we sort the activation channels according to their absolute mean values. Specifically, for the $k$th linear layer input tensor $\boldsymbol{X}^k \in \mathbb{R}^{L \times I}$, the channel-wise absolute mean vector $\boldsymbol{M}^k \in \mathbb{R}^I$ is computed as:

$$\boldsymbol{M}^k = \left( \frac{1}{L} \sum_{i=1}^{L} |X_{:,1}^k|, \frac{1}{L} \sum_{i=1}^{L} |X_{:,2}^k|, \ldots, \frac{1}{L} \sum_{i=1}^{L} |X_{:,I}^k| \right) \tag{9}$$

The permutation $\sigma^k$ is obtained by sorting the elements of $\boldsymbol{M}^k$ in ascending order. Let $p_4^k$, $p_6^k$, and $p_8^k$ denote the proportions obtained for $\boldsymbol{X}^k$; the channel partitioning is then defined as:

$$\boldsymbol{G}_4 = \sigma^k(\boldsymbol{X})_{:,:p_4^k I}, \quad \boldsymbol{G}_6 = \sigma^k(\boldsymbol{X})_{:,p_4^k I:(p_4^k+p_6^k)I}, \quad \boldsymbol{G}_8 = \sigma^k(\boldsymbol{X})_{:,(p_4^k+p_6^k)I:p_8^k I} \tag{10}$$

## 3.2 KERNEL DESIGN

Low-bit quantization offers significant performance improvements but presents considerable challenges in kernel design, especially for mixed-precision and fine-grained schemes like MicroMix. Recent advancements in GPU architectures, particularly the increased throughput of Tensor Cores for low-bit floating-point operations, combined with underlying support for block-scaled formats, have diminished the competitive advantage of traditional INT-type and GEMM quantization kernels. This simultaneously creates new opportunities for low-bit floating-point quantization.

**Mixed-precision Quantization.** Driven by the goal of deep algorithm-hardware integration, we have designed a mixed-precision, block-scaling quantization kernel for MicroMix. Complementing

this, we adopt MXFP-type GEMM kernels from CUTLASS, resulting in a kernel suite that delivers both excellent performance and high accuracy.

**Fine-grained Block-scaled Data Formats.** Quantization error is also influenced by the number of elements sharing a single scale factor. To fully harness the representational power of low-bit data types, fine-grained group quantization has become widely adopted and proven efficient in related works such as Atom and QuaRot. This used to be a tough trade-off between accuracy gains and dequantization overhead. However, the NVIDIA Blackwell architecture changes the game. Blackwell's mma instructions directly support new 4, 6, and 8-bit floating-point data types with integrated scale factors (known as MXFP formats), making block-scaled quantization a truly practical solution.

**GEMM Kernel.** As shown in Figure 4 (a), the output matrix is divided into blocks within each GEMM kernel, with iterations for each block performed along the $K$ dimension. After loading fragments of input matrices and their scale factors into Shared Memory or Tensor Memory, MMA instructions fused with dequantization operations are continuously executed on Tensor Cores. These operations accumulate FP32 partial sums into the BFloat16 result matrix. The process is highly decoupled, as matrices of specific data types invoke their corresponding GEMM kernels. This design allows for easy adjustment of data type categories and their ratios.

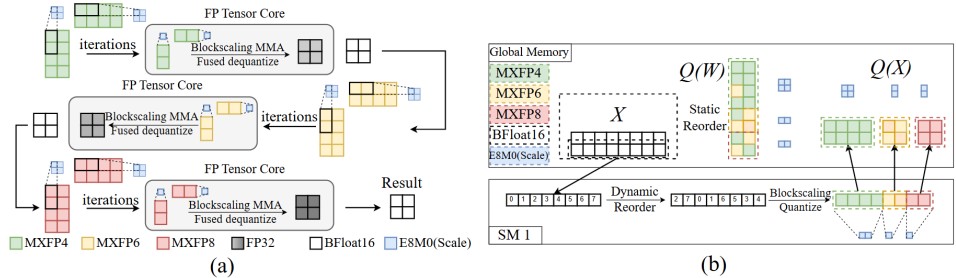

Figure 4: (a): The fused GEMM kernel of MicroMix. (b): The fused reorder-and-quantize operation. The quantization of weights is one-time cost and could be performed offline.

**Quantization Kernel.** Mixed-precision quantization often faces irregular memory access, leading to significant performance degradation. To tackle this, MicroMix adopts a strategy similar to Atom and RPTQ (Yuan et al., 2023) by reordering channels to enable regular memory access. Our algorithm divides channels of activation into three distinct parts, to which we then apply block-wise scaling quantization in 32-element blocks. To ensure correct matrix multiplication, weights are correspondingly permuted to match the reordered activations before undergoing a similar three-part block-wise scaling quantization. Crucially, the reordering and quantization of activations must occur dynamically, while these processes for weights can be handled offline as a pre-processing step. To mitigate the overhead of dynamic reordering, we employ a kernel fusion technique (see Figure 4 (b)), which combines the quantization and reordering operations into a single kernel. As shown in Figure 5, our fused kernel introduces little overhead compared to mixed-precision quantization only.

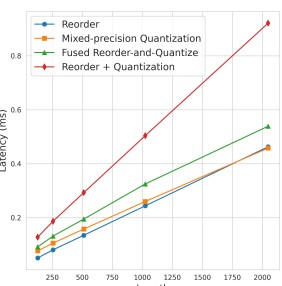

Figure 5: Comparison of the latency between single and fused operations with a batch size of 32.

# 4 EXPERIMENTS

## 4.1 EXPERIMENTAL SETUP

**Quantization.** MicroMix performs block-wise symmetric quantization with a block size of 32 for both weights and activations, using the E8M0 scaling format. The data formats are MXFP8 (E4M3), MXFP6 (E3M2), MXFP4 (E2M1) respectively. In Appendix D.1, Table 9 provides a summary of the quantized model information, including average using bits, offline calibration time and the size of quantized models.

**Baselines.** We compare MicroMix against four INT-based weight–activation quantization methods: Atom (Zhao et al., 2024), QUIK (Ashkboos et al., 2023), QuaRot (Ashkboos et al., 2024), FlatQuant (Sun et al., 2025) and one MX-based method, AMXFP4 (Lee et al., 2025). All baselines are reproduced on both Llama (Grattafiori et al., 2024) and Qwen (Qwen et al., 2025). Since MicroMix employs non-fixed bit-widths across linear layers, we additionally report the average bit-width per token element for all methods in Table 1. Implementation details are provided in Appendix D.1.

**Benchmarks.** For zero-shot evaluation, we use ARC_C (Clark et al., 2018), Lambada (Paperno et al., 2016), Winogrande (Sakaguchi et al., 2019), BoolQ (Clark et al., 2019), and PIQA (Lourie et al., 2021). For five-shot accuracy, we adopt MMLU (Hendrycks et al., 2021a). WikiText2 (Merity et al., 2016) is used to evaluate perplexity (PPL). Additionally, we assess the Code and Math capabilities of the Qwen2.5 model series. Code benchmarks are Human-Eval (Chen et al., 2021) and MBPP (Austin et al., 2021), while Math benchmarks cover GSM8K (Cobbe et al., 2021), MMLU-STEM (Hendrycks et al., 2021a), CMATH (Wei et al., 2023) and MATH (Hendrycks et al., 2021b).

## 4.2 MAIN RESULTS

Table 1 reports zero-shot and five-shot accuracy, along with WikiText-2 perplexity, for MicroMix and six baselines on Llama3.1-8B and Qwen2.5-32B. Across the five zero-shot benchmarks, MicroMix is the only quantization method that consistently preserves at least 98% of FP16 average accuracy on both models (Llama: 71.56 vs. 73.03; Qwen: 75.20 vs. 75.55). On the five-shot MMLU benchmark, MicroMix retains at least 96% of FP16 accuracy (Llama: 62.65 vs. 65.24; Qwen: 81.79 vs. 83.32), outperforming all competitors by $\geq$1.32 points on Llama and $\geq$0.27 points on Qwen. For WikiText2, MicroMix incurs only a marginal perplexity increase of 0.48 on Llama3.1-8B (6.72 vs. 6.24) and 0.46 on Qwen2.5-32B (5.56 vs. 5.02). The results of KV cache quantization is demonstrated in Table 10 of Appendix D.3.

Table 1: Zero-shot and few-shot accuracy and perplexity of Llama3.1-8B and Qwen2.5-32B evaluated with `lm-eval` (Gao et al., 2024). "Avg. Bits" denotes the average bit-width per token element. INT6 is implemented using symmetric per-token quantization.

| Model | Method | Avg. Bits | 0-shot (↑) | | | | | | 5-shot (↑) | PPL (↓) |
|---|---|---|---|---|---|---|---|---|---|---|
| | | | ARC_C | BoolQ | Lambada | PIQA | Winogrande | Avg. | MMLU | WikiText2 |
| Llama3.1-8B | FP16 | 16.00 | 53.58 | 81.99 | 75.47 | 80.09 | 74.03 | 73.03 | 65.24 | 6.24 |
| | QuaRot | 4.12 | 46.42 | 76.24 | 68.48 | 77.91 | 70.96 | 68.00 | 55.23 | 6.98 |
| | QUIK | 5.95 | 44.62 | 77.09 | 73.28 | 77.09 | 69.19 | 68.05 | 56.65 | 7.29 |
| | Atom | 4.25 | 50.17 | 76.15 | 69.45 | 78.02 | 70.01 | 68.76 | 58.05 | 6.79 |
| | FlatQuant | 4.19 | **51.54** | 78.87 | 73.32 | 79.16 | 71.98 | 70.97 | 61.33 | 6.95 |
| | AMXFP4 | 5.00 | 43.77 | 74.80 | 71.12 | 75.19 | 66.85 | 66.34 | 53.79 | 7.49 |
| | INT6 | 6.00 | 48.38 | 77.37 | 69.86 | 78.29 | 70.01 | 68.78 | 58.67 | 7.53 |
| | **MicroMix** | 5.51 | 50.26 | **81.13** | **74.13** | **80.14** | **72.14** | **71.56** | **62.65** | **6.72** |
| Qwen2.5-32B | FP16 | 16.00 | 55.89 | 87.46 | 76.21 | 82.26 | 75.93 | 75.55 | 83.32 | 5.02 |
| | QuaRot | 4.12 | 53.08 | 84.77 | 74.89 | **80.96** | 73.14 | 73.36 | 79.39 | 5.86 |
| | QUIK | 6.21 | 52.90 | 85.87 | 74.21 | 80.36 | 71.82 | 73.03 | 78.89 | 5.92 |
| | Atom | 4.21 | 54.78 | 86.54 | 75.92 | 81.45 | 73.48 | 74.43 | 79.54 | 5.89 |
| | FlatQuant | 4.71 | 56.23 | 86.30 | 75.41 | 81.50 | 74.19 | 74.72 | 81.52 | 5.74 |
| | AMXFP4 | 5.00 | 51.54 | 87.09 | 75.24 | 80.85 | 73.51 | 73.64 | 79.96 | 5.85 |
| | INT6 | 6.00 | 55.29 | 85.38 | 69.45 | 78.84 | 71.82 | 72.15 | 79.33 | 5.82 |
| | **MicroMix** | 5.22 | **56.66** | **87.13** | **77.37** | 80.65 | **74.19** | **75.20** | **81.79** | **5.56** |

Notably, higher average bit-width does not necessarily translate into higher accuracy. For instance, QUIK and INT6 employ more bits than MicroMix, yet provide limited performance gains.

Table 2: Mixtral-8x7B-v0.1-Instruct performance comparison between FP16 and MicroMix.

| | Arc_C | BoolQ | Lambada | PIQA | Winogrande | Avg. | Execution Time |
|---|---|---|---|---|---|---|---|
| FP16 | 65.70 | 88.50 | 77.37 | 84.49 | 76.87 | 78.58 | 5min 18s |
| MicroMix | 64.25 | 88.07 | 78.52 | 84.00 | 76.16 | 78.20 | 2min 03s |

As shown in Table 2, MicroMix attains accuracy comparable to FP16 on Mixtral-8x7B-v0.1-Instruct, with an average score drop of only 0.38 points (78.58 to 78.20) and per-task differences

typically within ±1.5 points. Notably, this minor accuracy trade-off comes with a substantial runtime reduction, cutting execution time from 5m18s to 2m03s.

**Math benchmarks.** Table 3 shows that MicroMix incurs an average accuracy drop of less than 4% compared to FP16, while retaining at least 98.4% of FP16 accuracy on GSM8K, MATH, and CMATH, with an average bit-width of 5.16.

Table 3: Accuracy (↑) of Qwen2.5-Math-7B-Instruct on math benchmarks: GSM8K, MMLU-STEM, CMATH, and MATH. FP8 is implemented by vLLM (Kwon et al., 2023).

| Model | Method | GSM8K | MATH | MMLU-STEM | CMATH | Average |
|---|---|---|---|---|---|---|
| 7B | FP16 | 95.8 | 83.7 | 77.8 | 91.5 | 87.2 |
| | FP8 | 95.5 | 83.4 | 68.7 | 91.7 | 84.8 |
| | MicroMix | 95.1 | 82.4 | 66.5 | 91.5 | 83.8 |

**Code benchmarks.** As reported in Table 4, MicroMix achieves accuracy comparable to or better than INT8 on the 14B (Avg. Bits: 5.54) and 32B (Avg. Bits: 5.18) models. Relative to FP16, the accuracy degradation remains within 1.5%.

Table 4: Accuracy (↑) of Qwen2.5-Coder-{14B,32B}-Instruct on Code benchmarks: Human-Eval and MBPP. INT8 is implemented by Bitsandbytes (Dettmers et al., 2022).

| Model | Method | Human-Eval | Human-Eval+ | MBPP | MBPP+ |
|---|---|---|---|---|---|
| 14B | FP16 | 87.8 | 84.1 | 81.0 | 69.2 |
| | INT8 | 86.6 | 82.9 | 86.0 | 73.0 |
| | MicroMix | 87.4 | 82.9 | 85.4 | 70.1 |
| 32B | FP16 | 88.4 | 84.1 | 84.5 | 70.9 |
| | INT8 | 89.0 | 85.4 | 86.2 | 73.5 |
| | MicroMix | 89.0 | 85.4 | 86.8 | 74.1 |

### 4.3 ABLATION STUDIES

In this section, we analyze the potential impact of different data formats and calibration datasets.

**MXFP6 and MXFP8 Variants.** We examine the impact of different MXFP6 (E2M3 and E3M2) and MXFP8 (E5M2 and E4M3) variants on zero-shot accuracy and perplexity. As shown in Table 5, all four configurations yield comparable results, suggesting that the specific exponent–mantissa trade-off has only a minor effect on MicroMix. This robustness arises because the definition of quantization thresholds explicitly accounts for the influence of both exponents and mantissas on quantization error, thereby mitigating sensitivity to data format choices.

Table 5: Zero-shot accuracy (↑) on Winogrande, Lambada, PIQA, and perplexity (↓) on WikiText2, using different exponent and mantissa bits for MXFP6 and MXFP8 on Llama3.1-8B. MXFP4 is E2M1 consistently.

| MXFP8 | MXFP6 | Winogrande | Lambada | PIQA | WikiText2 |
|---|---|---|---|---|---|
| E5M2 | E3M2 | 72.53 | 73.36 | **80.25** | 6.84 |
| | E2M3 | **72.69** | 72.83 | 80.14 | 6.81 |
| E4M3 | E3M2 | 72.14 | **74.13** | 80.14 | **6.72** |
| | E2M3 | 71.51 | 74.07 | 80.09 | 6.73 |

**Impact of Calibration Datasets.** To assess the robustness of offline partitioning, we test different calibration datasets, including WikiText2, Pile and C4 (Raffel et al., 2019). As shown in Table 12 of Appendix D.3, zero-shot and perplexity results remain stable across datasets with performance fluctuations within approximately 1%.

**Time Breakdown by Component.** We use the average values of $p_4$, $p_6$, and $p_8$ from Llama3.1-8B to compute the runtime breakdown of reorder-and-quantize and GEMM relative to the total

MicroMix kernel time, as shown in Table 6. Fused reorder-and-quantize operation takes less than 20% MicroMix kernel runtime across different lengths.

Table 6: The proportion of runtime for each part on RTX 5090.

| Part | Length = 128 | Length = 256 | Length = 512 | Length = 1024 | Length = 2048 | Length = 4096 |
|---|---|---|---|---|---|---|
| Reorder-and-Quantize | 7.9% | 9.7% | 11.8% | 14.3% | 17.0% | 16.9% |
| GEMM | 92.1% | 90.3% | 88.2% | 85.7% | 83.0% | 83.1% |

## 4.4 EFFICIENCY EVALUATION

In this section, we assess the efficiency of MicroMix from three perspectives: (1) single-kernel execution speed; (2) speedup of our custom kernel relative to CUTLASS; and (3) end-to-end performance in the prefill and decode stages. We evaluate MicroMix on three Blackwell architecture GPUs: RTX 5070Ti laptop, RTX 5090, and RTX PRO 6000, to examine its applicability across consumer and server GPUs.

**Kernel Efficiency.** We measure the latency of the MicroMix kernel across varying sequence lengths and hidden sizes. As baselines, we use strong TensorRT implementations: TensorRT FP8 (per tensor), W4A16 (per token), and FP16. Because MicroMix employs a nonfixed combination of 4-, 6-, and 8-bit channels, all kernel and transformer-block experiments report the minimum to maximum ranges alongside mean-value curves. As shown in Figure 6, MicroMix consistently outperforms TRT FP8 on both GPU platforms. On the RTX 5070Ti laptop, it achieves a 2.45 to 2.93× speedup over TRT FP16 and up to 1.45× over TRT FP8. On the RTX 5090, MicroMix delivers a 2.29 to 3.38× speedup over TRT FP16 and up to 1.74× over TRT FP8.

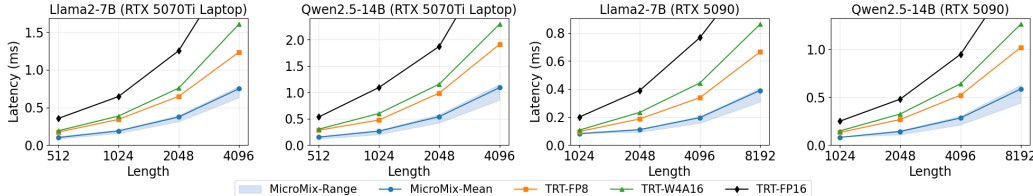

Figure 6: Computation latency of a single kernel with different lengths. "MicroMix-Range" denotes the latency span from the fastest to the slowest time.

**Performance of our customized kernel.** Table 7 reports the speedup of our customized GEMM kernel relative to CUTLASS. For small problem sizes with N = K = 4096, our kernels consistently outperform CUTLASS, achieving speedups of approximately 2.6 to 4.0× for W4A4, 3.1 to 5.0× for W6A6, and 1.2 to 3.1× for W8A8 as M increases. The gains are most pronounced at moderate M values, for example M = 32, indicating improved utilization and kernel efficiency for low-precision formats, particularly W6A6.

Table 7: Customized GEMM Kernel Speedup over CUTLASS on Small Problem Size (N=K=4096).

| M | W4A4 | | | W6A6 | | | W8A8 | | |
|---|---|---|---|---|---|---|---|---|---|
| | CUTLASS TFLOPS | Customized TFLOPS | Speedup | CUTLASS TFLOPS | Customized TFLOPS | Speedup | CUTLASS TFLOPS | Customized TFLOPS | Speedup |
| 1 | 1.04 | 2.72 | **2.62×** | 0.65 | 2.04 | **3.14×** | 1.02 | 1.63 | **1.60×** |
| 2 | 2.07 | 5.45 | **2.63×** | 1.30 | 4.67 | **3.59×** | 2.11 | 3.27 | **1.55×** |
| 4 | 4.16 | 10.91 | **2.62×** | 2.62 | 8.19 | **3.13×** | 4.16 | 6.55 | **1.57×** |
| 8 | 8.09 | 26.17 | **3.23×** | 5.23 | 18.72 | **3.58×** | 8.46 | 16.37 | **1.93×** |
| 16 | 15.95 | 52.37 | **3.28×** | 10.47 | 43.68 | **4.17×** | 17.01 | 37.42 | **2.20×** |
| 32 | 32.58 | 130.99 | **4.02×** | 20.95 | 104.75 | **5.00×** | 33.58 | 104.80 | **3.12×** |
| 64 | 66.82 | 209.50 | **3.14×** | 41.92 | 134.61 | **3.21×** | 67.28 | 130.96 | **1.95×** |
| 128 | 130.91 | 260.90 | **1.99×** | 83.83 | 161.35 | **1.92×** | 134.82 | 160.90 | **1.19×** |

**Comparison of 4-bit baselines.** To demonstrate the end-to-end efficiency of MicroMix, we compare against two 4-bit baselines, Atom and QuaRot. As shown in Figure 7, MicroMix reduces prefill

latency by about 85 percent compared to Atom and QuaRot on RTX PRO 6000. In the decode stage, MicroMix increases throughput by 1.82 to 3.02 times compared to Atom.

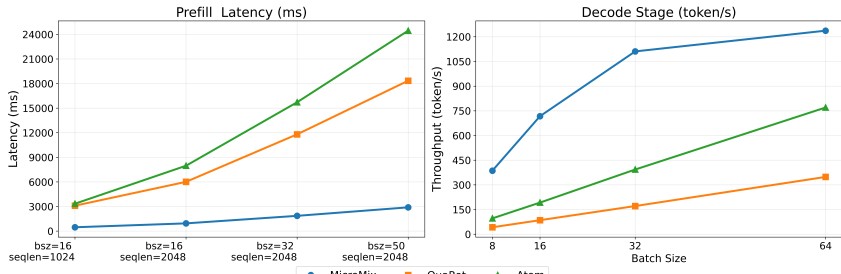

Figure 7: Prefill latency (left) and decoding throughput (right) of three methods on RTX PRO 6000.

**Comparison of FP16 and INT8 baselines.** On RTX 5090, we further compare the prefill performance of MicroMix against two baselines: FP16 from HuggingFace and INT8 from Bitsandbytes. Figure 8 reports the prefill latency and peak memory usage of Llama2-7B and Llama3.1-8B on the RTX 5090 with batch sizes $\{8, 12\}$ and sequence length 2048. Compared with FP16, MicroMix reduces memory usage by $2.29$–$2.84\times$ and latency by approximately $2.0\times$ at batch size 8. Compared with INT8, MicroMix further reduces memory usage by $1.60$–$2.01\times$ and latency by $1.80$–$1.84\times$ at batch size 12.

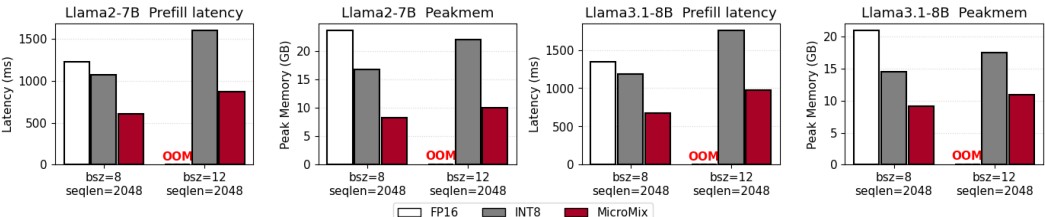

Figure 8: Prefill latency and peak memory usage of MicroMix compared with FP16 and INT8.

## 5 CONCLUSION

In this paper, we present MicroMix, a co-designed mixed-precision quantization algorithm and kernel that supports MXFP4, MXFP6, and MXFP8 formats. Our algorithm introduces the quantization threshold to identify elements that incur excessive quantization error at the target bit width. We also propose an offline calibration strategy to determine the optimal channel assignments for each precision level on calibration dataset. To enable efficient inference, we design a matrix multiplication kernel that integrates three GEMM precisions and a fused reorder-and-quantize operation. MicroMix kernel achieves significant speedups over TensorRT baselines on both RTX 5070Ti laptop and RTX 5090 GPUs across various configurations. On the RTX PRO 6000, MicroMix consistently outperforms FP16, INT8 and INT4 baselines.

## ACKNOWLEDGMENT

The authors would like to thank Yafei Zhao for his valuable contributions to the technical implementation.

This work is supported in part by the National Natural Science Foundation of China (Grant No. 62550068 and No. 62276188), and the Emerging Frontiers Cultivation Program of Tianjin University Interdisciplinary Center.

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

## A  RELATED WORKS

**Post-training Quantization** can be broadly divided into two categories: weight-only methods and weight–activation methods Lin et al. (2025a). Weight-only approaches (Frantar et al., 2023; Kim et al., 2024; Lin et al., 2024b) compress model weights into low-bit formats while dequantizing them back to high precision (e.g., FP16) during GEMM operations. Although this reduces memory bandwidth requirements, the computation itself still relies on high-precision operations, leaving a significant bottleneck in inference efficiency (Lin et al., 2025b). Consequently, there remains substantial room for accelerating LLM inference. Weight–activation methods (Yao et al., 2022; Lee et al., 2024; Lin et al., 2026), in contrast, quantize both weights and activations into low-bit formats, enabling GEMM to be executed entirely in low precision. This approach alleviates both bandwidth and computational bottlenecks but often suffers from severe accuracy degradation due to the presence of outlier activations. To address this challenge, mathematically equivalent transformation methods (Xiao et al., 2024; Shao et al., 2024) adopt a channel-level smoothing strategy. By shifting activation outliers into the weights, these methods effectively reduce quantization error. Rotation-based weight–activation methods (Ashkboos et al., 2024; Liu et al., 2024; Lin et al., 2024a) have recently emerged, achieving notable success in preserving model accuracy even at 4-bit precision.

**Mixed-precision quantization** retains outliers in higher bit-widths while quantizing the remaining elements to lower bit-widths (Dettmers et al., 2022; Saxena et al., 2025; Ashkboos et al., 2023; Hooper et al., 2025). The central challenge is designing efficient fused GEMM kernel. Atom (Zhao et al., 2024) achieves state-of-the-art performance by preserving 128 outlier channels in INT8 and quantizing the rest to INT4. Although Atom demonstrated a 7.73× speedup over FP16 on the RTX 4090, its current kernel is limited to Llama2-7B and can only handle up to 128 high-precision channels. Unlike previous approaches that use a fixed number of high-precision channels for all linear

layers, our method enables flexible, fine-grained mixed-precision configurations, and is specifically designed to leverage the advantages of Microscaling data formats.

**Applications of Microscaling data formats.** Recent works (Darvish Rouhani et al., 2023b; Sharify et al., 2024a;b) begin to study the applications of MX in both training and inference. AMXFP4 (Lee et al., 2025) handles outliers and asymmetries in activation by introducing asymmetric shared scales. Furthermore, Chen et al. (2025) significantly improved the FP4 training accuracy of Vision Transformers by identifying and solving the weight oscillation problem in forward propagation. MicroScopiQ (Akshat Ramachandran, 2025) optimizes the quantization by combining pruning with outlier-aware miniaturization. Although these works have made significant progress in the inference and training of low-width MX formats, there is still a lack of systematic work on using microscaling data formats for general mixed-precision quantization.

# B  MICROSCALING DATA FORMATS (MX)

According to Darvish Rouhani et al. (2023a), we give some supplementary information of MX in this section. An MX-compliant format is consisted of three components: scaling block size $k$, $k$ scalar elements $\{x_i\}_{i=1}^{k}$ and a shared scale $s$ in E8M0 format (see Figure 9). The special scale format enables the Microscaling data format to achieve dequantization operations solely through shift operations, thereby enhancing the running speed. Here, $\{x_i\}_{i=1}^{k}$ is already quantized, so the original value is $\{sx_i\}_{i=1}^{k}$. The specific parameters of MX data formats are shown in Table 8. More details on MX please refer to OCP Microscaling Specification (Darvish Rouhani et al., 2023a).

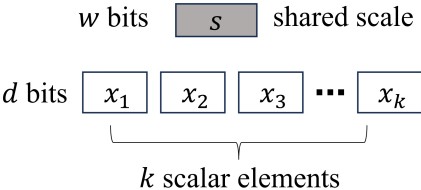

Figure 9: A schematic diagram of the basic unit of Microscaling block. The block encodes the original $k$ values $sx_i$ into $k$ elements in MX and a shared scale $s$.

Table 8: Format names and parameters of concrete MX-compliant formats (Darvish Rouhani et al., 2023a).

| Format Name | Element Bits ($d$) | Element Data Type | Exponent Bias ($b$) | Max Normal | Scaling Block Size ($k$) | Scale Data Type | Scale Bits ($w$) |
|---|---|---|---|---|---|---|---|
| MXFP8 | 8 | FP8 (E5M2) | 15 | $\pm 57344$ | 32 | E8M0 | 8 |
| | | FP8 (E4M3) | 7 | $\pm 448$ | | | |
| MXFP6 | 6 | FP6 (E3M2) | 3 | $\pm 28$ | 32 | E8M0 | 8 |
| | | FP6 (E2M3) | 1 | $\pm 7.5$ | | | |
| MXFP4 | 8 | FP4 (E2M1) | 1 | $\pm 6$ | 32 | E8M0 | 8 |
| MXINT8 | 8 | INT8 | N/A | $\pm 163/64$ | 32 | E8M0 | 8 |

# C  QUANTIZATION ERROR ANALYSIS

## C.1  OBSERVATIONS

In this section, we discuss the quantization error in detail. We observe the variation relationship between the accuracy of the quantization model and the quantization error, which supplements the deficiency in the description of the continuity relationship between accuracy and error in previous works.

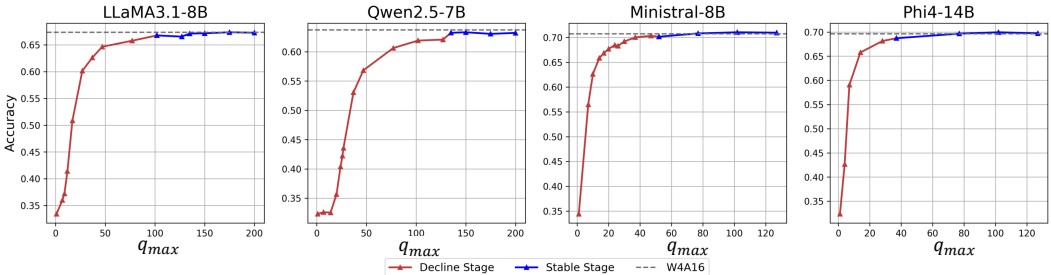

Figure 10: After quantizing weights to INT4 using per-channel symmetric quantization, the zero-shot average accuracy of the models on Winograde, PIQA, BoolQ, ARC_C, and Lambada changes with $q_{max}$. The quantization process of activations corresponding to different $q_{max}$ is implemented through fake-quant simulation. A lower value of $q_{max}$ corresponds to a higher upper bound on the quantization error.

Given a FP16 tensor $\boldsymbol{X} \in \mathbb{R}^{1 \times I}$, $\forall X_i \in \boldsymbol{X}$, the quantization error $E(X_i)$ between $Q(X_i)$ and $X_i$ is:

$$E(X_i) = |X_i - Q(X_i)| = |X_i - round(\frac{X_i}{s})s| = \gamma \cdot s \tag{11}$$

where $\gamma = |round(X_i/s) - X_i/s|$ is the rounding error. INT quantization is similar to FP quantization:

$$Q(X_i) = round(\frac{X_i}{s}), s = \frac{\max(|\boldsymbol{X}|)}{q_{max}} \tag{12}$$

where $round(\cdot)$ is rounding to the nearest INT value and $q_{max} = 2^{n-1} - 1$ is the maximum value of INT range. For INT format, there is $\gamma \in [0, 0.5]$, so we can get the quantization error upper bound $\overline{E}(X_i)$ of INT format:

$$\begin{aligned} E(X_i) = \gamma \cdot s &\leq 0.5 \cdot s \\ &= 0.5 \cdot \frac{\max(|\boldsymbol{X}|)}{2^{n-1} - 1} = \frac{\max(|\boldsymbol{X}|)}{2^n - 2} = \overline{E}(X_i) \end{aligned} \tag{13}$$

in particular, for INT8:

$$\overline{E}(X_i)_{INT8} = \frac{\max(|\boldsymbol{X}|)}{254} \tag{14}$$

We reformulate Equation 14 as following:

$$\overline{E}(X_i) = \frac{\max(|\boldsymbol{X}|)}{2 \cdot q_{max}} \tag{15}$$

Then we control $q_{max}$ to observe the relationship between the quantized model accuracy and the quantization error upper bound, as shown in Figure 10. We have three observations:

(1) The curve in Figure 10. clearly illustrates how model accuracy varies with quantization error. In general, the accuracy of the model decreases with the increase of the upper bound of the quantization error.

(2) There is a "Stable Stage" for each model maintaining high accuracy of variation $q_{max}$, INT8 ($q_{max}$=127) is located in this stage. For all four models, INT8 is a high-precision format.

(3) When $q_{max}$ is below a threshold, the accuracy of quantized model degrades significantly, which we name as "Decline Stage", and INT4 ($q_{max}$=7) is located at the end of this stage.

In conclusion, enhancing the accuracy of a quantized model requires reducing its quantization error to bring it within the stable stage. The relationship between the quantization error upper bound and the model accuracy inspires us to divide values into three parts from the view of quantization error upper bound.

## C.2 Derivations

In this section, we show the detailed derivation processes of quantization threshold, which is based on the motivation of controlling the quantization error of MXFP4/MXFP6 below $\overline{E}(X)_{INT8}$. The quantization error of MXFP4/MXFP6 is:

$$E(X_i)_{\{MXFP4,MXFP6\}} = \gamma \cdot 2^{\lfloor \log_2(\max(|\boldsymbol{X}|)) \rfloor - b} \tag{16}$$

Since the gap between adjacent FP values is not a constant, we use

$$\gamma = \frac{q_{max}}{2^{n-1}} \tag{17}$$

to approximately express the rounding error in Equation 16, where $q_{max}$ is the maximum value of MXFP4/MXFP6. Substituting Equation 17 into Equation 16 gives:

$$
\begin{aligned}
E(X_i) &= \frac{q_{max}}{2^{n-1}} \cdot 2^{\lfloor \log_2(\max(|\boldsymbol{X}|)) \rfloor - b} \\
&\leq \frac{q_{max}}{2^{n-1}} \cdot 2^{\log_2(\max(|\boldsymbol{X}|)) - b} \\
&= \frac{q_{max}}{2^{n-1}} \cdot \frac{\max(|\boldsymbol{X}|)}{2^b}
\end{aligned}
\tag{18}
$$

Let $E(X_i)_{\{MXFP4,MXFP6\}} \leq \overline{E}(X_i)_{INT8}$. Then we have

$$E(X_i) \leq \frac{q_{max}}{2^{n-1}} \cdot \frac{\max(|\boldsymbol{G}_{\{4,6\}}|)}{2^b} \leq \overline{E}(X_i)_{INT8} \tag{19}$$

If the inequality on the right-hand side holds, it follows that

$$\max(|\boldsymbol{G}_{\{4,6\}}|) \leq 2^b \cdot \frac{2^{n-1}}{q_{max}} \cdot \frac{\max(|\boldsymbol{X}|)}{254} \tag{20}$$

According to Table 8, when $n = 4$ or $n = 6$, the corresponding values of $q_{\max}$ and $b$ can be substituted directly into Equation 20. At last, we get the definition of quantization threshold:

$$T(n) = 2^b \cdot \frac{2^{n-1}}{q_{max}} \cdot \frac{\max(|\boldsymbol{X}|)}{254} \tag{21}$$

# D Supplementary Materials of Experiments

## D.1 Experimental Settings

In this section, we demonstrate some reproduction details, especially claiming how "Avg.Bits" in Table 1 is calculated.

**QuaRot**[1]. QuaRot uses symmetric INT4 quantization of group size 128. $a\_clip\_ratio$ is 0.9, and $w\_clip$ is used. For QuaRot, its online Hadamard transformation depends on `Fast_Hadamard_Transform`[2] kernel without introducing extra matrices. So its average bits is:

$$4 + \frac{1}{128} \cdot 16 = 4.12 \tag{22}$$

**Atom**[3]. The activation-sort metric is chosen as "hessian" according to Atom's default settings. $a\_clip\_ratio$ is 0.9, $w\_clip\_ratio$ is 0.85 and $keeper\_size$ is 128. The "Avg.Bits" of Atom is calculated as following:

$$\frac{((hidden\_size - 128) \cdot 4 + 128 \cdot 8 + \frac{hidden\_size}{128} \cdot 16}{hidden\_size} \tag{23}$$

---

[1]https://github.com/spcl/QuaRot/tree/main

[2]https://github.com/Dao-AILab/fast-hadamard-transform

[3]https://github.com/efeslab/Atom

**QUIK**[4]**.** The value of $fp\_features\_num$ is set to 256, following the settings used in QUIK. The part of INT4 is quantized using asymmetric per-token quantization. $w\_clip$ and $int8\_down\_proj$ is used. Since QUIK adopts pure INT8 for all Down Projs and mixed-precision for other linear layers, its "Avg. Bits" is:

$$\frac{((hidden\_size - 256) \cdot 4 + 256 \cdot 16 + 2 \cdot 16) \cdot 6 + (intermediate\_size \cdot 8 + 16)}{hidden\_size \cdot 6 + intermediate\_size} \tag{24}$$

where 6 counts for Q, K, V, O, Up and Gate Projs.

**AMXFP4**[5]**.** We use the $fp4\_e2m1\_asym$ element format as specified in AMXFP4. $scale\_bits$ is 8 and $block\_size$ is 32. $scale\_mode = 2$ (default setting in $run.sh$) needs two FP16 scales for each block, so its "Avg. Bits" is:

$$\frac{32 \cdot 4 + 2 \cdot 16}{32} = 5 \tag{25}$$

**FlatQuant**[6]**.** We adopt the per-token and per-channel INT4 symmetric quantization for FlatQuant, with parameters such as $lwc$, $lac$, $cali\_trans$, and $add\_diag$. Since FlatQuant introduces 7 additional square transformation matrices per layer during the forward pass, and the elements of these matrices differ across decoder layers, its "Avg. Bits" is computed as follows:

$$\frac{(hidden\_size \cdot 6 + intermediate\_size) \cdot 4 \cdot seqlen + numel(\boldsymbol{P}) \cdot 16}{(hidden\_size \cdot 6 + intermediate\_size) \cdot seqlen} \tag{26}$$

where $numel(\boldsymbol{P})$ denotes the sum of the elements of the transformation matrices per layer:

$$numel(\boldsymbol{P}) = \begin{cases} 64 \cdot 64 \cdot 4 + 32 \cdot 32 + 112 \cdot 112 + 128 \cdot 128 = 46336, \text{Llama 3.1-8B} \\ 64 \cdot 64 \cdot 2 + 80 \cdot 80 \cdot 2 + 40 \cdot 40 + 144 \cdot 144 + 192 \cdot 192 = 80192, \text{Qwen 2.5-32B} \end{cases} \tag{27}$$

Since the additional introduced matrix can be reused for all tokens, when the seqlen is longer, the average number of bits caused by transformation matrices is lower. When $seqlen > 190$, the average number of additional bits introduced by $\boldsymbol{P}$ is less than 0.01. But at the same time, the single-token situation in the decode stage has to be taken into consideration. In conclusion, we uniformly set $seqlen = 100$.

### D.2 Information on Post-Quantization Models

Table 9 shows some information of quantized models. In general, MicroMix utilizes 5-5.6 bits for Llama and Qwen series models. The offline calibration and quantization time are relatively fast, which only takes 2min23s to get the quantized model of Qwen2.5-Math-7B-Instruct.

Table 9: Average bit-width per element and memory consumption of quantized weights across all evaluated models. "Quantization Time" denotes the total offline time cost, including reordering and quantization of the original model weights.

| Models | Avg. Bits | Memory | Quantization Time |
|---|---|---|---|
| Llama3.1-8B | 5.51 | 5.09 GB | 179s |
| Qwen2.5-32B | 5.22 | 24.54 GB | 406s |
| Qwen2.5-Coder-14B-Instruct | 5.54 | 9.10 GB | 260s |
| Qwen2.5-Coder-32B-Instruct | 5.18 | 24.53 GB | 406s |
| Qwen2.5-Math-7B-Instruct | 5.16 | 4.79 GB | 143s |

In Figure 3, we tallied $p_4$, $p_6$ and $p_8$ of each layer of Llama3.1-8B. We supplement the average number of bits for each layer in Figure 11.

Figure 12 illustrates the precision mapping in MicroMix. The reorder-and-quantize operation is fused into LayerNorm, and the resulting quantized activations are reused by subsequent linear layers. In addition, the KV cache is quantized with FlashInfer (Ye et al., 2024) to further reduce memory usage.

---

[4]https://github.com/IST-DASLab/QUIK

[5]https://github.com/aiha-lab/MX-QLLM

[6]github.com/ruikangliu/FlatQuant

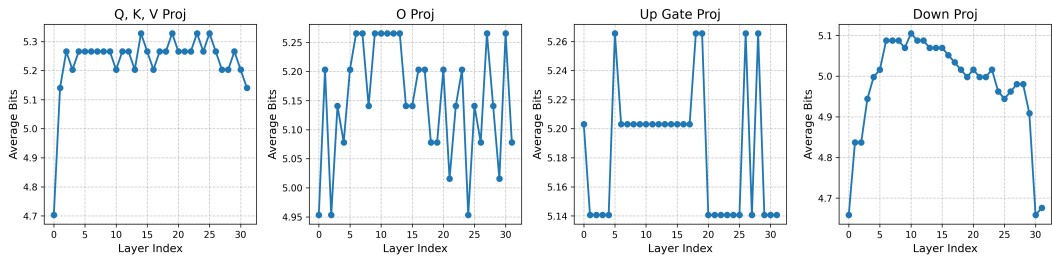

Figure 11: The average number of bits per layer of Llama3.1-8B.

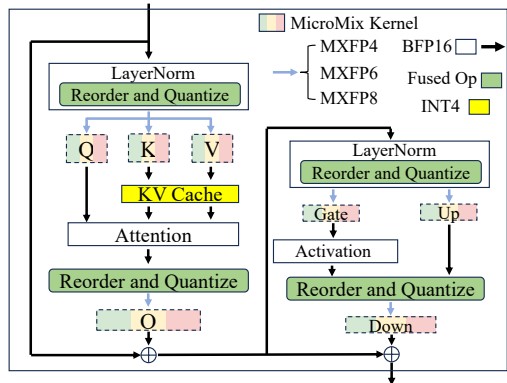

Figure 12: Precision mapping of MicroMix for a Transformer block in LLM.

## D.3 SUPPLEMENTARY RESULTS

The results of KV cache quantization are reported in Table 10, where all methods adopt INT4 asymmetric quantization with a group size of 64. MicroMix retains over 97.6% of the FP16 zero-shot accuracy under this setting. For five-shot accuracy and perplexity, MicroMix also achieves state-of-the-art performance.

Table 10: Zero-shot, few-shot accuracy and perplexity of Llama3.1-8B, using lm-eval (Gao et al., 2024). All methods use asymmetric INT4 quantization of group size 64.

| Model | Method | Avg. Bits | 0-shot (↑) | | | | | | 5-shot (↑) | PPL (↓) |
|---|---|---|---|---|---|---|---|---|---|---|
| | | | ARC_C | BoolQ | Lambada | PIQA | Winogrande | Avg. | MMLU | WikiText2 |
| | FP16 | 16.00 | 53.58 | 81.99 | 75.47 | 80.09 | 74.03 | 73.03 | 65.24 | 6.24 |
| Llama3.1-8B | QuaRot | 4.12 | 44.88 | 74.19 | 66.74 | 77.09 | 66.61 | 65.90 | 53.19 | 8.03 |
| | QUIK | 5.95 | 49.66 | 77.77 | 71.08 | 78.51 | 67.88 | 68.98 | 57.63 | 7.32 |
| | Atom | 4.25 | 47.95 | 79.94 | 72.81 | 78.35 | 70.72 | 69.95 | 57.10 | 7.43 |
| | FlatQuant | 4.19 | 50.17 | 79.33 | 72.15 | 79.27 | 71.59 | 70.50 | 59.34 | 7.12 |
| | AMXFP4 | 5.00 | 46.84 | 73.24 | 69.59 | 77.15 | 67.56 | 66.87 | 53.11 | 7.33 |
| | INT6 | 6.00 | 49.15 | 76.73 | 68.62 | 78.07 | 69.06 | 68.32 | 56.82 | 7.82 |
| | **MicroMix** | 5.51 | **52.22** | **80.64** | **74.00** | **79.38** | **70.96** | **71.44** | **60.90** | **6.97** |

Table 11 shows the results of Atom and QUIK directly applied to MXFP4 and MXFP8, with a significant performance drop compared to MicroMix. Since the kernels of Atom and QUIK do not support the MXFP format, we use the MicroMix kernel to keep the number of MXFP8 channels at 128 and 256 respectively.

We further test the stability and flexibility of MicroMix to adapt different average bit-widths (4, 5 and 6), as shown in Table 13.

Table 11: Zero-shot accuracy (↑) and WikiText2 perplexity (↓) results of mixed-precision methods on MXFP formats using Llama3.1-8B.

| Methods | ARC_C | BoolQ | Lambada | PIQA | WikiText2 |
|---------|-------|-------|---------|------|-----------|
| FP16 | 51.28 | 82.05 | 75.80 | 80.03 | 6.24 |
| Atom | 43.60 | 76.36 | 66.52 | 75.57 | 8.02 |
| QUIK | 47.27 | 76.15 | 68.52 | 76.72 | 7.86 |
| MicroMix | **50.17** | **81.13** | **74.13** | **80.14** | **6.72** |

Table 12: Zero-shot accuracy (↑) on ARC_C, BoolQ, Lambada, PIQA, and perplexity (↓) on Wiki-Text2 of MicroMix on different datasets using Qwen2.5-14B.

| Calib Data | ARC_C | BoolQ | Lambada | PIQA | WikiText2 |
|------------|-------|-------|---------|------|-----------|
| WikiText2 | 57.59 | 86.18 | 74.29 | 81.12 | 5.87 |
| Pile | 57.68 | 85.41 | 74.13 | 80.14 | 5.92 |
| C4 | 58.36 | 86.57 | 73.65 | 81.56 | 5.92 |

Table 13: Using Qwen2.5-32B, we report the accuracy and performance of MicroMix on various average bits.

| Avg Bits | Arc_C | BoolQ | Lambada | PIQA | Winogrande | Avg. | Execution Time |
|----------|-------|-------|---------|------|------------|------|----------------|
| 4.48 | 56.91 | 85.87 | 76.54 | 81.28 | 73.88 | 74.89 | 5min 37s |
| 4.84 | 55.46 | 86.51 | 77.18 | 81.50 | 74.19 | 74.96 | 5min 42s |
| 5.09 | 56.14 | 85.66 | 77.14 | 81.39 | 74.98 | 75.06 | 5min 44s |
| 5.22 (ours) | 56.66 | 87.13 | 77.37 | 80.65 | 74.19 | 75.20 | 5min 46s |
| 5.30 | 55.38 | 86.09 | 77.24 | 81.66 | 75.06 | 75.08 | 5min 48s |
| 5.74 | 55.63 | 86.02 | 76.65 | 81.77 | 74.19 | 74.85 | 5min 49s |
| 6.01 | 55.78 | 86.27 | 76.67 | 81.72 | 74.66 | 75.02 | 5min 50s |

