# OpenReview forum: "MicroMix: Efficient Mixed-Precision Quantization with Microscaling Formats for Large Language Models"
_ICLR.cc/2026/Conference — ICLR 2026 Poster_

### Official Review · Reviewer_BbmQ · 2025-10-27

**Soundness:** 2
**Presentation:** 3
**Contribution:** 2
**Rating:** 6
**Confidence:** 3

**Summary:**

This paper proposes MicroMix, a novel mixed-precision quantization approach that leverages the MXFP4, MXFP6, and MXFP8 formats supported by recent NVIDIA architectures. The method performs channel-wise mixed-precision quantization for both weights and activations, assigning different bit-widths based on value distribution thresholds. In addition to the quantization algorithm, the paper presents a system-level co-design, including fused kernels for quantization, permutation, and GEMM operations to minimize latency and memory overhead. Extensive experiments on large language models (Llama3.1-8B and so on) demonstrate that MicroMix achieves state-of-the-art performance without retraining. The work highlights both algorithmic insights and system-level engineering optimizations for efficient mixed-precision inference on modern GPUs.

**Strengths:**

The method adopts a channel-wise mixed-precision strategy, which is more flexible and effective than conventional layer-wise approaches.

The insight for permutation and assign different bit-widths through threshold is conceptually clean yet empirically powerful.

The work tightly couples algorithmic design with low-level kernel fusion

Extensive experiments have been conducted to prove the effectiveness of MicroMix for both accuracy and system efficiency.

**Weaknesses:**

The work is more engineering-oriented than fundamentally novel from a research standpoint.

**Questions:**

I doubt the permutation and mixed-precision channels multiplication may cause extra latency.

In Figure 7, what exactly does “end-to-end throughput” measure?

---

> ### Author Response · Authors · 2025-11-18
>
> Thank you for the positive and constructive feedback. Below we provide detailed responses to your concerns.
>
> **W1**: The work is more engineering-oriented than fundamentally novel from a research standpoint.
>
> **Answer**:
> Thank you for pointing out your concerns about our theoretical innovation. The starting point of our research is to define the quantization threshold for microscaling data formats (MX). Based on the threshold, clearly distinguish those elements that cause excessive quantization errors in the MXFP4/MXFP6 format, and then there will be corresponding engineering practices.
> We further explain the academic significance of quantization threshold of MX as follows:
>
> - **MX-specific thresholding.** Existing outlier definitions, for example LLM.int8() treating $|x|$ > 6 as outliers and DuQuant using $|x|$ > 1000 for extreme outliers, are independent of the target bit-width or numeric format (INT vs. FP). In contrast, we derive a closed-form, format-aware threshold $T(n)$ that depends on MX parameters (exponent bias $b$, $q_{max}$) and the target bit-width $n$.
>
>
> - **Necessity of MX-tailored thresholds.** Table 9 (applying Atom and QUIK directly to microscaling formats) shows that transplanting mixed-precision methods based on legacy outlier definitions to MX induces substantial accuracy degradation, demonstrating the necessity of defining quantization thresholds specifically for microscaling data formats.
>
> - **Lack of a precise mathematical criterion for activation quantizability.** Prior approaches such as SmoothQuant and QuaRot argue effectiveness by showing that smoothed or rotated matrices “appear” more uniform, which is a heuristic distribution-level observation lacking a strict mathematical test. Our threshold transforms the decision of whether to raise bit-width from heuristics into a computable inequality, $\max(|G_n|) \leq T(n)$, which directly pins low-bit errors within the INT8 stability envelope. This criterion naturally adapts with bit-width and MX format parameters, and aligns with the empirically identified stability region (our Fig. 9).
>
> **Q1**: I doubt the permutation and mixed-precision channels multiplication may cause extra latency.
>
> **Answer**:
> After analysis using Nsight Systems, we find that the permutation-quantization kernel accounts for only about 1% of the total end-to-end time. And permutation brings approximately 25% overhead to the permutation-quantization kernel. So the overhead brought by permutataion (1%*25%=0.25%) is basically negligible.
>
> After optimization, MicroMix achieves an average kernel launch time of about 15 microseconds during decoding, compared to roughly 30 microseconds for the FP16 GEMM kernel. This indicates that mixed-precision utilization introduces minimal overhead.
>
> **Q2**: In Figure 7, what exactly does “end-to-end throughput” measure?
>
> **Answer**: Figure 7 reports Prefill_tokens/Prefill_time. We provide decode performance of MicroMix compared to 4-bit baselines on an RTX PRO 6000:
>
> ### Llama-2-7B Decode Performance (token/s, prefill length=256)
> | Method | bsz=8 | bsz=16 | bsz=32 | bsz=64 |
> |-|-|-|-|-|
> | MicroMix | 386 | 717 | 1111 | 1237 | 1315 |
> | QuaRot | 42 | 85 | 171 | 348 | 689 |
> | Atom | 96 | 192 | 393 | 770 |
>
> QuaRot employs the CUTLASS-based implementation as well.
> Atom (INT4+INT8) has not yet been optimized for the Blackwell architecture, which limits its runtime performance in experiments on RTX PRO6000. As noted in QServe, per-group integer quantization requires an integer-to-floating-point dequantization for partial sums, introducing additional overhead. This contributes to the observed slowdown. These limitations motivate our design of an FP mixed-precision quantization algorithm, which avoids the per-group integer dequantization bottleneck and achieves faster execution on Blackwell.

---

> > ### Comment · Reviewer_BbmQ · 2025-11-28
> > **Response to the authors**
> >
> > Thank you for your response. I choose to keep my rating.

---

### Official Review · Reviewer_6987 · 2025-10-31

**Soundness:** 2
**Presentation:** 2
**Contribution:** 2
**Rating:** 4
**Confidence:** 4

**Summary:**

This paper proposes MicroMix, a mixed-precision quantization framework designed for Microscaling (MX) floating-point formats (MXFP4, MXFP6, MXFP8), targeting NVIDIA’s Blackwell FP4 Tensor Cores.
It introduces:

A quantization threshold–based precision assignment, determining which channels use 4-, 6-, or 8-bit MX formats.

An offline calibration to precompute per-layer precision ratios.

A fused reorder-and-quantize kernel optimized for MX formats.

MicroMix reportedly achieves near-FP16 accuracy at ~5 bits average precision and up to 3.3× kernel-level speedup on RTX 5090, outperforming FP16 and INT8 baselines.

**Strengths:**

The paper targets a relevant and timely topic, leveraging upcoming FP4/6/8 Tensor Cores for efficient LLM inference.

The system–algorithm co-design is well-executed, integrating quantization thresholds and fused CUDA kernels.

**Weaknesses:**

The main algorithmic ideas—mixed-precision assignment, threshold-based outlier handling, and fused reorder kernels—are incremental extensions of prior works. The combination feels straightforward rather than conceptually new.

The method mostly refines known strategies (precision partitioning, block scaling) for a specific hardware (Blackwell FP4), rather than introducing a new quantization principle.

**Questions:**

The paper is competently executed and well-written, with solid experiments and system results.
However, the conceptual novelty is limited—it mostly integrates existing techniques (mixed precision + threshold + MX formats + kernel fusion) rather than proposing a fundamentally new approach. Could you clearly articulate what aspect of MicroMix is genuinely novel compared to prior mixed-precision or microscaling quantization methods?

How sensitive is the threshold-based precision assignment to the choice of calibration data and ratio parameters?

Could the proposed mixed-precision calibration generalize to other quantization formats (e.g., MXINT4/8,NVFP4)?

---

> ### Author Response · Authors · 2025-11-18
>
> Thank you for your careful review and constructive suggestions. We will combine experiments to answer your doubts.
>
> **W1**: The main algorithmic ideas—mixed-precision assignment, threshold-based outlier handling, and fused reorder kernels—are incremental extensions of prior works. The combination feels straightforward rather than conceptually new.
>
> **Q1**: The paper is competently executed and well-written, with solid experiments and system results. Could you clearly articulate what aspect of MicroMix is genuinely novel compared to prior mixed-precision or microscaling quantization methods?
>
> **Answer**: To address the concerns of algorithm novelty, we discuss the difference of MicroMix from prior mixed-precision quantization methods as following two points:
> - **MX-specific thresholding.** Existing outlier definitions, for example LLM.int8() treating $|x|$ > 6 as outliers and DuQuant using $|x|$ > 1000 for extreme outliers, are independent of the target bit-width or numeric format (INT vs. FP). In contrast, we derive a closed-form, format-aware threshold $T(n)$ that depends on MX parameters (exponent bias $b$, $q_{max}$) and the target bit-width $n$.
>
> - **Necessity of MX-tailored thresholds.** Table 9 (applying Atom and QUIK directly to microscaling formats) shows that transplanting mixed-precision methods based on legacy outlier definitions to MX induces substantial accuracy degradation, demonstrating the necessity of defining quantization thresholds specifically for microscaling data formats.
>
>
> **W2**: The method mostly refines known strategies (precision partitioning, block scaling) for a specific hardware (Blackwell FP4), rather than introducing a new quantization principle.
>
> **Answer**: Indeed, we agree that some techniques used in MicroMix, like precision partitioning and block scaling, are capable with Blackwell architecture and not new quantization principles. We treat it as the problem of trade-off between practicality and novelty.
>
>
> **Q2**: How sensitive is the threshold-based precision assignment to the choice of calibration data and ratio parameters?
> Regarding the robustness to calibration data, we provide a detailed discussion in Sec. 4.3 Ablation Studies (Table 10). The conclusion is that different calibration datasets affect accuracy within ±1% and perplexity within ±0.1.
>
> Regarding the robustness of ratio parameters, we revisit the paper’s definition of the quantization threshold $T$:
> $$
> T(n) = 2^b · \frac{2^{(n−1)}}{q_{max}}\cdot \frac{max(|X|)}{254}
> $$
>
> This definition contains no hyperparameters. The terms $b$ and $q_{max}$ vary but are uniquely determined by the target format. As for MicroMix’s robustness across different target formats (MXFP6-E2M3, MXFP6-E3M2, MXFP8-E4M3, MXFP8-E5M2), Sec. 4.3 Ablation Studies (Table 4) presents detailed results, showing accuracy variations within ±0.8% and perplexity variations within ±0.1.
>
> We infer that “ratio parameters” may refer to different average bit budgets. Below are supplementary results on Qwen2.5-32B for MicroMix under varying average bits:
>
> | Avg Bits | Arc_C  | BoolQ  | Lambada | PIQA   | Winogrande | Avg.     | Execution Time |
> |-|-|-|-|-|-|-|--|
> | 4.48     | 56.91  | 85.87  | 76.54   | 81.28  | 73.88      | 74.89   | 5min 37s       |
> | 4.84     | 55.46  | 86.51  | 77.18   | 81.5   | 74.19      | 74.96   | 5min 42s       |
> | 5.09     | 56.14  | 85.66  | 77.14   | 81.39  | 74.98      | 75.06   | 5min 44s       |
> | 5.22 (ours) |56.66|	87.13 | 77.37| 80.65|74.19|	75.20| 5min46s
> | 5.3      | 55.38  | 86.09  | 77.24   | 81.66  | 75.06      | 75.08   | 5min 48s       |
> | 5.74     | 55.63  | 86.02  | 76.65   | 81.77  | 74.19      | 74.85   | 5min 49s       |
> | 6.01     | 55.78  | 86.27  | 76.67   | 81.72  | 74.66      | 75.02   | 5min 50s       |
>
> It can be seen that the accuracy of MicroMix remains stable in cases with an average bit size of 4.x, demonstrating stability to adapt to a lower bit width. However, under our setting (5.22), when the number of bits continues to increase, the accuracy benefit begins to decrease, while the latency cost still grows.

---

> ### Author Response · Authors · 2025-11-18
>
> **Q3**: Could the proposed mixed-precision calibration generalize to other quantization formats (e.g., MXINT4/8,NVFP4)?
>
> Your insight is correct: MicroMix can be extended to other data formats such as MXINT4/8 and NVFP4. We illustrate this with MXINT4/8.
>
> First, MXINT4/8 adopts block-wise quantization with a block size of 32, where all elements in a block share an E8M0 scale.
> Accordingly, the quantization error of MXINT4 (as in Eq. 16 of the paper) can be written as:
>
> $$
> E(X\_i)\_{MXINT4}=\gamma\cdot 2^{\lfloor \log\_2(\max(|\boldsymbol{X}|))\rfloor }
> $$
>
> Imposing $E(X\_i)\_{MXINT4}\leq\overline{E}(X\_i)\_{INT8}$ yields:
>
> $$
> \max(|\boldsymbol{G}\_{4}|)\leq\frac{2^4-2}{2^8-2} \cdot \max(|\boldsymbol{X}|)
> $$
>
> The results of MXINT4/MXINT8 on zero-shot benchmarks:
>
> | Method | Avg Bits | Arc_C | BoolQ | Lambada | PIQA | Winogrande | Avg. |
> |--------|----------|-------|-------|---------|------|------------|------|
> | MicroMix-MXFP4/6/8 | 4.48 | 56.91 | 85.87 | 76.54 | 81.28 | 73.88 | 74.89 |
> | MicroMix-MXFP4/6/8 | 4.84 | 55.46 | 86.51 | 77.18 | 81.5 | 74.19 | 74.96 |
> | MicroMix-MXINT4/8 | 4.5 | 56.23 | 86.18 | 76.67 | 81.7 | 74.15 | 74.98 |
>
> MicroMix also performs well with MXINT4/8.

---

### Official Review · Reviewer_MRQT · 2025-11-01

**Soundness:** 3
**Presentation:** 2
**Contribution:** 2
**Rating:** 4
**Confidence:** 3

**Summary:**

This paper presents MicroMix, a mixed MXFP8/FP6/FP4 quantization method that mainly targets NVIDIA's Blackwell generation GPU and benefits from the new FP4 tensor cores. A quantization algorithm is proposed to allocate different bitwidths to different channels, and a fused quantization kernel is implemented to support the dynamic reordering and mixed precision quantization. The performance is evaluated on consumer-grade GPUs (RTX5070Ti laptop & RTX5090), and it demonstrates a 2.29-3.38x speedup compared to the FP16 baseline.

**Strengths:**

It is good to see a work that benefits from the new technologies on a cutting-edge GPU and demonstrates end-to-end gains and memory savings.

**Weaknesses:**

- The mixed precision schema partitions the weight+activation into MXFP8+MXFP8, MXFP6+MXFP6, and MXFP4+MXFP4. Among them, MXFP6+MXFP6 does not make sense to me. I believe FP6 has the same peak performance on Blackwell GPUs (correct me if I'm wrong), and quantizing **activations** into FP6 does not reduce the model size, nor increase the inference speed. W6A8 might be a better choice here, and cutlass has an example for W6A8 kernel (https://github.com/NVIDIA/cutlass/blob/main/examples/79_blackwell_geforce_gemm/79c_blackwell_geforce_mixed_mxfp8_mxfp6_bf16_gemm.cu)

- The real efficiency of the implemented kernel is not very clear to me. Although there is a comparison against TRT-FP16, W4A16, and FP8 in section 4.4, the actual utilization (achieved vs peak performance) is not provided.

- Though this paper claims they conduct experiments on server-grade GPUs (RTX5090), the RTX5090 still has the same architecture as consumer GPUs (sm_120) instead of real compute cards like B200 with sm_100 architecture. It is totally understandable that the authors may not have access to these latest-generation GPUs, but I would not recommend claiming the work supports server-grade GPUs. Also, some explanations in the paper (for example, tcgen05.mma instructions) only apply to these compute cards and do not exist on RTX5090.

**Questions:**

- What is the hardware utilization of the mixed precision kernel? I would recommend using the ratio of FP4/FP6/FP8 and the peak performance numbers to calculate the theoretical latency and compare it with the measured latency. (Be aware that on GeForce GPUs, the bitwidth of accumulation (FP32 vs FP16) will affect the peak performance.) I'm concerned about this because MicroMix uses 3 cutlass kernels to perform a single GEMM, and this might lead to some overhead.

- In end-to-end throughput results, why does INT8 quantization provide little performance improvement? Also, what are the performances of the prefilling stage (TTFT) and decoding stage (token/s)?

- I would recommend adding some accuracy+performance results in the same table. How do quantization methods in Table 1 perform?

---

> ### Author Response · Authors · 2025-11-18
>
> Thank you for your insightful feedback. We highly value these suggestions and are pleased to provide further clarifications and discussions below.
>
> **W1**: W6A8 might be a better choice instead of W6A6.
>
> **Answer**: Thank you for the suggestion! Indeed, W6A8 delivers clearly superior performance compared to W6A6 in our measurements.  However, we used W6A6 in our deployment studies to stay aligned with the algorithmic design: our quantization thresholding partitions activations such that $T(4) < G_6 ≤ T(6)$, and elements in $G_6$ can be represented with 6 bits while keeping quantization error low, without needing escalation to 8 bits.
> A reasonable compromise is to use W6A6 for accuracy evaluation, reflecting the academic value of the threshold design, and switch to W6A8 for deployment-time throughput benchmarking to further improve MicroMix’s end-to-end performance. This approach preserves algorithmic consistency while also delivering stronger engineering results.
>
> **W2** and **Q1**: About GEMM kernel efficiency.
>
> **Answer**:
> We understand your concerns and have further supplemented the experiment:
> ### CUTLASS GEMM Kernel Performance
> | Kernel | Achieved TFLOPS | Peak TFLOPS (dense) | % of Peak |
> |-|-|-|-|
> | W4A4   | 1268.01          | 1676                 | 75.66     |
> | W6A6   | 392.56           | 838                  | 46.84     |
> | W8A8   | 529.88           | 838                  | 63.23     |
> | W6A8   | 589.62           | 838                  | 70.36     |
>
> We have made optimizations for small problem size:
> |M| W4A4 CUTLASS TFLOPS | W4A4 Customized TFLOPS | W4A4 Speedup | W6A6 CUTLASS TFLOPS | W6A6 Customized TFLOPS | W6A6 Speedup | W8A8 CUTLASS TFLOPS | W8A8 Customized TFLOPS | W8A8 Speedup |
> |-|-|-|-|-|-|-|-|-|-|
> |1|1.04|2.72|**2.62x**|0.65|2.04|**3.14x**|1.02|1.63|**1.60x**|
> |2|2.07|5.45|**2.63x**|1.30|4.67| **3.59x**|2.11|3.27|**1.55x**       |
> |4|4.16|10.91|**2.62x**|2.62|8.19|**3.13x**|4.16|6.55| **1.57x**|
> |8|8.09|26.17|**3.23x**|5.23|18.72|**3.58x**|8.46|16.37| **1.93x**|
> |16|15.95|52.37|**3.28x**|10.47|43.68|**4.17x**|17.01|37.42|**2.20x**|
> |32|32.58|130.99|**4.02x**|20.95|104.75|**5.00x**|33.58|104.80|**3.12x**|
> |64|66.82|209.50|**3.14x**|41.92| 134.61|**3.21x**|67.28|130.96|**1.95x**|
> |128|130.91|260.90|**1.99x**|83.83|161.35|**1.92x**|134.82|160.90| **1.19x**|
>
> The performance of all three kernels is improved to varying degrees.
>
> **W3** argues that despite the paper’s claim of server-grade GPU experiments, using an RTX 5090 (consumer sm_120) rather than true compute cards like B200 (sm_100) undermines that claim, and some provided explanations (e.g., tcgen05.mma) only apply to those compute GPUs and not the RTX 5090.
>
> **Answer**:
> Thank you for your careful reading and understanding! Indeed, due to our limited resources, we are unable to debug on the B200. We will modify the expressions related to the sm_100 architecture. We provide the results on RTX PRO6000 in the answer of **Q2** to verify that MicroMix can directly run on server-grade GPU.
>
> **Q2**: In end-to-end throughput results, why does INT8 quantization provide little performance improvement? Also, what are the performances of the prefilling stage (TTFT) and decoding stage (token/s)?
>
> **Answer**:
> The INT8 used in this paper is derived from the Bitsandbytes library. We speculate that the reason for the relatively small speed improvement is its use of the mixed-precision method of LLM.int8().
>
> More end-to-end results are as follows:
> ### Llama-2-7B Prefill Latency (ms)
> | Method | bsz=16 seqlen=1024 | bsz=16 seqlen=2048 | bsz=32 seqlen=2048 | bsz=50 seqlen=2048 |
> |-|-|-|-|-|
> | MicroMix | 463 | 938 | 1857 | 2893 |
> | QuaRot | 3094 | 6005 | 11801 | 18341 |
> | Atom | 3334 | 7970 | 15714 | 24451 |
>
> ### Llama-2-7B Decode Performance (token/s, prefill length=256)
> | Method | bsz=8 | bsz=16 | bsz=32 | bsz=64 |
> |-|-|-|-|-|
> | MicroMix | 386 | 717 | 1111 | 1237 | 1315 |
> | QuaRot | 42 | 85 | 171 | 348 | 689 |
> | Atom | 96 | 192 | 393 | 770 |
>
> QuaRot employs the CUTLASS-based implementation as well.
> Atom (INT4+INT8) has not yet been optimized for the Blackwell architecture, which limits its runtime performance in experiments on RTX PRO6000. As noted in QServe, per-group integer quantization requires an integer-to-floating-point dequantization for partial sums, introducing additional overhead. This contributes to the observed slowdown. These limitations motivate our design of an FP mixed-precision quantization algorithm, which avoids the per-group integer dequantization bottleneck and achieves faster execution on Blackwell.

---

> ### Author Response · Authors · 2025-11-18
>
> **Q3**: I would recommend adding some accuracy+performance results in the same table. How do quantization methods in Table 1 perform?
>
> **Answer**:
> It is a good suggestion to display both the accuracy and the execution time simultaneously in Table 1. However, the code of [QuaRot]( https://github.com/spcl/QuaRot/tree/main/fake_quant),  [Atom](https://github.com/efeslab/Atom/blob/main/model/quant.py),  [AMXFP4]( https://github.com/aiha-lab/MX-QLLM/blob/main/microxcaling/mx/cpp/mx.cuh) and [FlatQuant](https://github.com/ruikangliu/FlatQuant/blob/main/flatquant/quant_utils.py) use **fake quant** instead of real quantization during accuracy evaluation. That is, using float16 to represent the quantized data to simulate the quantization effect, it could not reflect the real running speed of their method on benchmarks.
>
> Here we add a result for the MOE model Mixtral-8x7B-Instruct v0.1 (on RTX PRO 6000):
> |        | Arc_C  | BoolQ  | Lambada | PIQA   | Winogrande | Avg.    | Execution Time |
> |-|-|-|-|-|-|-|-|
> | FP16   | 65.70   | 88.50   | 77.37   | 84.49  | 76.87      | 78.58  | 5min18s        |
> | MicroMix| 64.25 | 88.07  | 78.52   | 84.00     | 76.16      | 78.20    | 2min03s        |
>
> With an average of 4.74 bits used, MicroMix's zero-shot average accuracy has only decreased by **0.38%** compared to the original model!

---

### Official Review · Reviewer_x7qv · 2025-11-01

**Soundness:** 2
**Presentation:** 2
**Contribution:** 2
**Rating:** 4
**Confidence:** 3

**Summary:**

This paper proposes a mixed-precision post-training quantization framework using MX formats with 4, 6, and 8-bit channels. The method adaptively assigns precision to channels based on quantization thresholds and fuses channel reordering into the quantization kernel. The approach is designed to leverage NVIDIA Blackwell FP4 tensor cores. Experiments on Llama and Qwen models show better accuracy with approximately five-bit effective precision and latency improvements.

**Strengths:**

The authors provide an end-to-end implementation and report real latency improvements on Blackwell GPUs, which increases the paper's practical value.

The use of flexible bit-width allocation per layer and integration with FP4 tensor core support matches new hardware capabilities. It is timely.

The empirical results show high accuracy retention across zero-shot, math, and code benchmarks. The fused reorder-and-quantize kernel design is efficient and demonstrates careful engineering.

**Weaknesses:**

Despite the solid engineering contribution, the technical novelty appears limited. The paper extends prior mixed-precision ideas, such as Atom, by allowing flexible channel ratios. The threshold-based assignment and layer-wise granularity resemble existing mixed precision heuristics and tuning methods, so the algorithmic insight is incremental.  Fusing permutation into quantization kernel is primarily an engineering effort.

The end-to-end speedup analysis uses FP16 and FP8 baselines, but not the low-bit kernels like 4bit. A comparison against an equivalent or similar bit-width kernel would clarify relative benefits. Separating prefilling and decoding latency would also give better insights. The evaluation could extend to more settings of different batch sizes, context lengths and decoding lengths. Right now only a few settings are covered. The evaluated models are only Llama and Qwen language models.

Figure 6 compares kernel speedups against FP8, FP16, and W4A16, but it remains unclear how much of the speed improvement arises from the authors’ design versus underlying CUTLASS kernels. A breakdown of kernel time contributions would clarify the engineering contribution.

For the accuracy comparison, the proposed method MicroMix is only set to ~5 bit. However, there are baselines like QuaRot use ~4bit.  It would be fair to also show ~4bit result for MicroMix, as well as for the latency comparison.

**Questions:**

Rather than only showing results around five bit width, could you please analyze results at different target bit widths such as approximately four, five, or six bits to demonstrate the flexibility of the method and its ability to adapt to different accuracy and latency budgets?

Could you please analyze more model architecture like different attention modules and MoE etc to show its generality?

---

> ### Author Response · Authors · 2025-11-18
>
> We appreciate your careful review and the suggestion to supplement experiments concerning the average number of bits used. Detailed responses and results are provided below.
>
> **W1**: Despite the solid engineering contribution, the technical novelty appears limited. The paper extends prior mixed-precision ideas, such as Atom, by allowing flexible channel ratios. The threshold-based assignment and layer-wise granularity resemble existing mixed precision heuristics and tuning methods, so the algorithmic insight is incremental.
>
> **Answer**:
> Thank you for acknowledging our engineering contributions. Regarding technical novelty, we agree that certain components of MicroMix, such as threshold-based assignment and fine-grained quantization, are similar to prior mixed-precision approaches. These techniques have been validated by related work as both meaningful and practical.
>
> However, Table 9 in Appendix E.3, which directly applies Atom and QUIK to microscaling formats, shows that transplanting mixed-precision methods built on previous outlier definitions into MX leads to substantial accuracy degradation. This finding underscores the necessity of defining quantization thresholds specifically tailored to microscaling data formats.
>
>
>
> **Q1**: Rather than only showing results around five bit width, could you please analyze results at different target bit widths such as approximately four, five, or six bits to demonstrate the flexibility of the method and its ability to adapt to different accuracy and latency budgets?
>
> **Answer**: Below we provide supplementary experiments on Qwen2.5-32B addressing different cases.
>
> | Avg Bits | Arc_C  | BoolQ  | Lambada | PIQA   | Winogrande | Avg.     | Execution Time |
> |-|-|-|-|-|-|-|--|
> | 4.48     | 56.91  | 85.87  | 76.54   | 81.28  | 73.88      | 74.89   | 5min 37s       |
> | 4.84     | 55.46  | 86.51  | 77.18   | 81.5   | 74.19      | 74.96   | 5min 42s       |
> | 5.09     | 56.14  | 85.66  | 77.14   | 81.39  | 74.98      | 75.06   | 5min 44s       |
> | 5.22 (ours) |56.66|	87.13 | 77.37| 80.65|74.19|	75.20| 5min 46s
> | 5.3      | 55.38  | 86.09  | 77.24   | 81.66  | 75.06      | 75.08   | 5min 48s       |
> | 5.74     | 55.63  | 86.02  | 76.65   | 81.77  | 74.19      | 74.85   | 5min 49s       |
> | 6.01     | 55.78  | 86.27  | 76.67   | 81.72  | 74.66      | 75.02   | 5min 50s       |
>
> It can be seen that the accuracy of MicroMix remains stable in cases with an average bit size of 4.x, demonstrating stability to adapt to a lower bit width. However, under our setting (5.22), when the number of bits continues to increase, the accuracy benefit begins to decrease, while the latency cost still grows.
>
> **W2**: The end-to-end speedup analysis uses FP16 and FP8 baselines, but not the low-bit kernels like 4bit.
>
> **W4**: For the accuracy comparison, the proposed method MicroMix is only set to ~5 bit. However, there are baselines like QuaRot use ~4bit. It would be fair to also show ~4bit result for MicroMix, as well as for the latency comparison.
>
> **Answer**: To address your concerns, we supplement the comparison experiments with QuaRot and Atom at 4.x bits. The results of zero-shot benchmarks:
>
> |        | Avg Bits | Arc_C  | BoolQ  | Lambada | PIQA   | Winogrande | Avg.     |
> |-|-|-|-|-|-|-|-|
> | QuaRot | 4.12     | 53.08  | 84.77  | 74.89   | 80.96  | 73.14      | 73.36   |
> | Atom   | 4.25     | 54.78  | 86.54  | 75.92   | 81.45  | 73.48      | 74.43  |
> | MicroMix | 4.48   | 56.91  | 85.87  | 76.54   | 81.28  | 73.88      | 74.89  |
> | MicroMix | 4.84   | 55.46  | 86.51  | 77.18   | 81.5   | 74.19      | 74.96   |
>
> *Since the average bit count of pure MXFP4 is 4.25(4+8/32), MXFP4/6/8 cannot be very close to 4 bits.
>
> The results of Prefill stage and Decode stage:
> ### Llama-2-7B Prefill Latency (ms)
> | Method | bsz=16 seqlen=1024 | bsz=16 seqlen=2048 | bsz=32 seqlen=2048 | bsz=50 seqlen=2048 |
> |-|-|-|-|-|
> | MicroMix | 463 | 938 | 1857 | 2893 |
> | QuaRot | 3094 | 6005 | 11801 | 18341 |
> | Atom | 3334 | 7970 | 15714 | 24451 |
>
> ### Llama-2-7B Decode Performance (token/s, prefill length=256)
> | Method | bsz=8 | bsz=16 | bsz=32 | bsz=64 | bsz=128 |
> |-|-|-|-|-|-|
> | MicroMix-ours | 386 | 717 | 1111 | 1237 | 1315 |
> | QuaRot | 42 | 85 | 171 | 348 | 689 |
> | Atom | 96 | 192 | 393 | 770 | 1464 |
>
> QuaRot employs the CUTLASS-based implementation as well. Atom (INT4+INT8) has not yet been optimized for the Blackwell architecture, which limits its runtime performance in experiments on RTX PRO6000. As noted in QServe, per-group integer quantization requires an integer-to-floating-point dequantization for partial sums, introducing additional overhead. This contributes to the observed slowdown. These limitations motivate our design of an FP mixed-precision quantization algorithm, which avoids the per-group integer dequantization bottleneck and achieves faster execution on Blackwell.

---

> ### Author Response · Authors · 2025-11-18
>
> **W3**: It remains unclear how much of the speed improvement arises from the authors’ design versus underlying CUTLASS kernels.
>
> **Answer**:
> Thank you for your suggestion. We have made optimizations for small problem size:
> ### Customized GEMM Kernel Speedup over CUTLASS on Small Problem Size (N=K=4096)
>
> |M| W4A4 CUTLASS TFLOPS | W4A4 Customized TFLOPS | W4A4 Speedup | W6A6 CUTLASS TFLOPS | W6A6 Customized TFLOPS | W6A6 Speedup | W8A8 CUTLASS TFLOPS | W8A8 Customized TFLOPS | W8A8 Speedup |
> |-|-|-|-|-|-|-|-|-|-|
> |1|1.04|2.72|**2.62x**|0.65|2.04|**3.14x**|1.02|1.63|**1.60x**|
> |2|2.07|5.45|**2.63x**|1.30|4.67| **3.59x**|2.11|3.27|**1.55x**       |
> |4|4.16|10.91|**2.62x**|2.62|8.19|**3.13x**|4.16|6.55| **1.57x**|
> |8|8.09|26.17|**3.23x**|5.23|18.72|**3.58x**|8.46|16.37| **1.93x**|
> |16|15.95|52.37|**3.28x**|10.47|43.68|**4.17x**|17.01|37.42|**2.20x**|
> |32|32.58|130.99|**4.02x**|20.95|104.75|**5.00x**|33.58|104.80|**3.12x**|
> |64|66.82|209.50|**3.14x**|41.92| 134.61|**3.21x**|67.28|130.96|**1.95x**|
> |128|130.91|260.90|**1.99x**|83.83|161.35|**1.92x**|134.82|160.90| **1.19x**|
>
> The performance of all three kernels is improved to varying degrees.
>
> **Q2**: Could you please analyze more model architecture like different attention modules and MoE etc to show its generality?
>
> **Answer**: Here we add a result for the MOE model Mixtral-8x7B-Instruct v0.1 (on RTX PRO 6000):
> |        | Arc_C  | BoolQ  | Lambada | PIQA   | Winogrande | Avg.    | Execution Time |
> |-|-|-|-|-|-|-|-|
> | FP16   | 65.70   | 88.50   | 77.37   | 84.49  | 76.87      | 78.58  | 5min18s        |
> | MicroMix| 64.25 | 88.07  | 78.52   | 84.00     | 76.16      | 78.20    | 2min03s        |
>
> With an average of 4.74 bits used, MicroMix's zero-shot average accuracy has only decreased by **0.38%** compared to the original model!

---

### Author Response · Authors · 2025-12-03
**General Response**

We sincerely thank all four reviewers for their insightful feedback.

We are encouraged that all reviewers acknowledged our contribution in introducing a novel and advanced mixed-precision method based on microscaling data formats, particularly our solid engineering contributions on Blackwell GPUs.

Reviewers x7qv, 6987, and BbmQ raised concerns about the theoretical novelty. In response, our paper introduces a quantization threshold under microscaling data formats that explicitly identifies elements causing excessive quantization error at a given target bit-width, together with a precise mathematical characterization of quantization difficulty, thereby bridging theory and practice in a verifiable way.

All four reviewers requested additional experiments, including:
- Extending MicroMix to the MoE model Mixtral-8x7B-Instruct v0.1.
- Budget analysis of accuracy–efficiency trade-offs under varying average bit-widths.
- Prefill and decode stage efficiency comparisons against 4-bit baselines.
- Efficiency comparison between our customized kernel and the original CUTLASS kernel.

All the requested experiments have been integrated into the revised manuscript. Below, we provide detailed point-by-point responses to each reviewer’s comments.

---

### Meta-Review · Area_Chair_wXry · 2025-12-27

**Summary:**

This paper proposes MicroMix, a mixed-precision quantization framework based on Microscaling (MX) formats, designed to maximize the performance of FP4 Tensor Cores in NVIDIA’s Blackwell architecture. While reviewers generally praised the paper's hardware optimization and high level of engineering completeness, they raised significant concerns regarding technical novelty. Specifically, they pointed out that the core ideas—such as mixed-precision and channel reordering—seem to be incremental improvements over existing works like Atom. Additionally, concerns were raised regarding the validity of claiming server-grade performance based on consumer GPU (RTX 5090) results and the lack of validation across diverse settings, such as the 4-bit bandwidth range.

**Reviewer Concerns:**

**Addressed Concerns**
 - Lack of Experimental Diversity: In the rebuttal, the authors successfully addressed concerns about the method's generalization capabilities by adding results for the MoE model (Mixtral-8x7B). They demonstrated a 2.5x speedup with only a negligible 0.38% accuracy loss compared to FP16. Furthermore, they provided comparisons with 4-bit baselines (QuaRot, Atom) and analyzed performance across various average bit-widths (4 to 6 bits), proving the flexibility and competitiveness of their approach.
 - Kernel Efficiency Verification: The authors demonstrated performance advantages over CUTLASS even for small problem sizes. They also presented data showing that the reordering overhead is negligible (approximately 0.25% of the total execution time), effectively resolving doubts regarding system efficiency.

**Outstanding Concerns**
 - Novelty and Datacenter GPU Verification: Reviewers still rate the academic novelty as low, viewing the work primarily as tuning existing methodologies for MX formats. Additionally, due to the lack of access to actual Blackwell Datacenter GPUs (e.g., B200), doubts regarding actual utilization in server-grade environments have not been fully dispelled.

**Reviewer Scores:**

The reviewers assigned scores of 6, 4, 4, and 4. Regrettably, only the reviewer who assigned the score of 6 (Reviewer BbmQ) responded to the rebuttal, while the reviewers who assigned scores of 4 (x7qv, MRQT, 6987) did not participate in the post-rebuttal discussion.
From the AC's perspective, I acknowledge that the fundamental academic novelty of this work is inherently limited, as it builds upon existing mixed-precision frameworks. Furthermore, there is a valid practical limitation regarding the absence of experiments on high-end datacenter GPUs, such as the NVIDIA B200, which constrains the verification of server-grade performance claims.

However, despite these limitations, the paper deserves credit for proposing an "advanced" mixed-precision methodology specifically tailored for MXFP formats. The authors provided a sound mathematical approach to derive quantization thresholds, effectively bridging the gap between theoretical error analysis and hardware constraints. I believe this yields a degree of specific novelty that distinguishes it from a mere engineering port. Had the reviewers engaged in a fuller discussion regarding this mathematical rigor and the timely application to emerging formats, there was a possibility for the scores to be adjusted upward.

---

### Decision · Program_Chairs · 2026-01-26

Accept (Poster)